# A new methodology to train fracture network simulation

# using Multiple Point Statistics

Pierre-Olivier BRUNA[(1)*], Julien STRAUBHAAR[(2)], Rahul PRABHAKARAN[(1, 3)], Giovanni

BERTOTTI[(1)], Kevin BISDOM[(4)], Grégoire MARIETHOZ[(5)], Marco MEDA[(6)].

(1) Department of Geoscience and Engineering, Delft University of Technology, Delft, the Netherlands.
(2) Centre d'hydrogéologie et de géothermie (CHYN), Université de Neuchâtel, Emile-Argand 11, CH-2000 Neuchâtel.
(3) Department of Mechanical Engineering, Section of Energy Technology, Eindhoven University of Technology, Eindhoven the Netherlands.
(4) Shell Global Solutions International B.V., Grasweg 31, 1031HW Amsterdam, The Netherlands
(5) University of Lausanne, Institute of Earth Surface Dynamics (IDYST) UNIL-Mouline, Geopolis, office 3337, 1015 Lausanne, Switzerland
(6) ENI Spa, Upstream and Technical Services, San Donato Milanese, Italy.

* Corresponding author, p.b.r.bruna@tudelft.nl

Keywords: geostatistics, multiple training images, probability map, fracture networks, stress-induced fracture aperture, outcrop.

## Abstract

Natural fracture network characteristics can be known from high-resolution outcrop images acquired from drone and photogrammetry. Such images might also be good analogues of subsurface naturally fractured reservoirs and can be used to make predictions of the fracture geometry and efficiency at depth. However, even when supplementing fractured reservoir models with outcrop data, gaps in that model will remain and fracture network extrapolation methods are required. In this paper we used fracture networks interpreted in two outcrops from the Apodi area in Brazil to present a revised and innovative method of fracture network geometry prediction using the Multiple Point Statistics (MPS) method.

The MPS method presented in this article uses a series of small synthetic training images
(TIs) representing the geological variability of fracture parameters observed locally in the
field. The TIs contain the statistical characteristics of the network (i.e. orientation, spacing,
length/height and topology) and allow representing complex arrangement of fracture
networks. These images are flexible as they can be simply sketched by the user.
We proposed to use simultaneously a set of training images in specific elementary zones of
the Apodi outcrops in order to best replicate the non-stationarity of the reference network. A
sensitivity analysis was conducted to emphasize the influence of the conditioning data, the
simulation parameters and the used training images. Fracture density computations were
performed on selected realisations and compared to the reference outcrop fracture
interpretation to qualitatively evaluate the accuracy of our simulations. The method proposed
here is adaptable in terms of training images and probability map to ensure the geological
complexity is accounted for in the simulation process. It can be used on any type of rock
containing natural fractures in any kind of tectonic context. This workflow can also be applied
to the subsurface to predict the fracture arrangement and fluid flow efficiency in water,
geothermal or hydrocarbon fractured reservoirs.
**I] Introduction**
**I.1 The importance of the prediction of fracture network geometry**
Fractures are widespread in Nature and depending on their density and their aperture, they
might have a strong impact on fluid flow and fluid aquifers (Berkowitz, 2002; Rzonca, 2008),
and in geothermal (Montanari et al., 2017; Wang et al., 2016) and hydrocarbon reservoirs
(Agar and Geiger, 2015; Lamarche et al., 2017; Solano et al., 2010) They are typically
organised as networks ranging from nanometre to multi-kilometre scale (Zhang, 2016), and
present systematic geometrical characteristics (i.e. type, orientation, size, topology) that are
determined from specific stress and strain conditions. These conditions have been used to
derive concepts of fracture arrangements in various tectonic contexts and introduced the
notion of geological fracture-drivers (fault, fold, burial, facies). Based on these drivers it is
possible to some extent to predict reservoir heterogeneity and to define potential permeability
pathways within the rock mass (Lamarche et al., 2017; Laubach et al., 2018). Despite the
existence of these concepts, a range of parameters including fracture abutment relationships as
well as height/length distributions cannot be adequately sampled along a 1D borehole and are
mainly invisible on seismic images. In addition, fracture networks may present a spatial
complexity (variability of orientation or clustering effect) that is also largely unknown in the
subsurface. Long and Witherspoon, (1985) and Olson et al., (2009) showed how those
parameters impact the connectivity of the network and consequently affect fluid flow in the
subsurface. In outcrops, the fracture network characteristics can be observed in 2D and
understood directly. Consequently, outcrops are essential to characterize fracture network
attributes that cannot be sampled in the subsurface, such as length or spatial connectivity.

**I.2 Surface rocks as multiscale reservoir analogues**
In this context, the study of outcrop analogues is one of the few ways to constrain the
architecture of fracture networks (Bisdom et al., 2014; Bruna et al., 2017; National Research
Council, 1996; Lamarche et al., 2012; Lavenu et al., 2013). Outcrops can be considered as a
natural laboratory where the structural reality can be observed and quantified at various
scales. At the small – measurement station – scale (order of 10's m), fracture type,
chronologies and topology relationships can be characterised using classical ground-based
structural geology method such as scanlines (Lavenu et al., 2013; Mauldon et al., 2001). At
the intermediate – outcrop – scale (order of $10^2$'s m), length of fractures and geometry
variability can be qualified and quantified using unmanned aerial vehicles (UAV - drones).
Working on outcrops allows an understanding of the geological history of the targeted area
and possibly to decipher how, when and where fractures were developed. In addition,
outcrops constitute an efficient experimental laboratory where some of properties of the
fracture network (i.e. fracture distribution, apertures, permeability and fluid flow behaviour)
can be known and modelled (Bisdom et al., 2017). At the large − reservoir − scale (order of
$10^{3-4}$m) satellite imagery and geophysical maps provide the characterisation of the 100's of
meter long objects such as large fracture systems or faults.
However, not every outcrop can be considered as a good analogue for the subsurface. Li et al.,
(2018), in their work on the Upper Cretaceous Frontier Formation reservoir, USA, observed
significant differences in the fracture network arrangement in subsurface cores compared to
an apparent good surface analogue of the studied reservoir. In the subsurface, fractures
appeared more clustered than in the outcrop where the arrangement is undistinguishable from
random. The origin of these differences is still debated but these authors suggest that
alteration (diagenesis) or local change in pressure-temperature conditions, may have
contributed to the observed variability. The near-surface alteration processes (exhumation,
weathering) may also contribute to misinterpretations of the characteristics of the network. In
this case, one should be particularly careful while using observed networks to make geometry
or efficiency (porosity, permeability) predictions in the subsurface. Therefore, the application
to the subsurface of the characteristics observed in the outcrop is not always straightforward
or even possible, and may lead to erroneous interpretations. Relatively unbiased signals such
as stylolites or veins and particular geometric patterns build trust that the studied outcrop can
be compared to the subsurface.

**I.3 Modelling approaches classically used to model fracture network geometries**
The widely used discrete fracture network (DFN) stochastic modelling tools provide
statistical representation of fracture networks constrained generally by univariate and random
distribution of orientation, size, spacing and density/intensity data (Bisdom et al., 2014;
Bisdom et al., 2017; Huang et al., 2017; Panza et al., 2018). The generated models follow a
local stationarity hypothesis. This implies that the statistics used during the simulation are
constant in the defined area of interest (Deutsch and Journel, 1997; Gringarten and Deutsch,
1999; Gringarten and Deutsch, 2001; Journel and Zhang, 2006). Liu et al., (2009), highlighted
the implicit randomisation that conventional DFN models produce and demonstrated that
parameters like fracture connectivity are poorly considered in these representations. In
addition, it is generally admitted that discrete realisations of thousands of fracture objects at
the kilometre scale are computationally very demanding and often even impossible (Jung et
al., 2013). Some authors attempted to use a pixel-based method to try to predict fracture
network geometries. Bruna et al., (2015), used a dense hydrogeological borehole survey
sampling a Lower Cretaceous aquifer in the SE of France to define fracture facies and to
model their distribution with two-points geostatistics. In this case, the amount of available
data and their consistency helped to provide realistic results. However, far from conditioning
data (i.e. boreholes) the fractures simulation are poorly constrained.
The work of Hanke et al., (2018) uses a directional semi-variogram to quantify fracture
intensity variability and intersection density. This contribution provides an interesting way to
evaluate the outputs of classical DFN approaches but requires a large quantity of input data
that are not always available in the subsurface. To represent the fracture network geometry in
various geological contexts, an alternative method has to be developed. This innovative
method needs to i) explicitly predict the organisation and the characteristics of multiscale
fracture objects, ii) take into consideration the spatial variability of the network and iii)
require a limited amount of data to be realised.
**I.4 Multi-point statistics as an alternative to classic DFN approaches**
Since Liu et al., (2002), few authors highlighted the potential of using multi-point statistics
(MPS) to generate realistic fracture networks (Chugunova et al., 2017; Karimpouli et al.,
2017). Strebelle, (2002) showed how the MPS are able to reproduce any type of geological
heterogeneities of any shape at any size as long as they present a repetitive character. This
characteristic seems particularly well adapted to predict the geometry of a fracture network.
The MPS method uses training images (TI) to integrate conceptual geological knowledge into
geostatistical simulations (Mariethoz, 2009). The TI is a grid containing geological patterns
that are representative of a certain type of geological structure, type and arrangement. The TI
can be considered as a synthetic model of the geological heterogeneity (i.e. all the elements
characterising a geological object) likely to occur in a larger domain (i.e. reservoir, aquifer,
outcrop). The TI must contain the range of geobodies that are intended to be modelled, as well
as the relationship these geobodies have with each other (Mariethoz, 2009; Strebelle, 2002).

**I.5 Objectives and contents of this research**
In this paper we propose a MPS workflow considering the geological variability of the
fracture network geometry in outcrops (size order of 100m) and a methodology on how to use
this method at the reservoir scale. The approach is based on the direct sampling method
(Mariethoz et al., 2010) and uses multiple TIs for a single realisation (Wu et al., 2008). The
concept of the probability map has been revised here to define where a training image should
be used in the simulation grid. Our outcrop-based simulations also take into account "seismic-
scale" objects (i.e. object longer than 40m) considered as hard conditioning data. The
proposed workflow is tested on outcrops where fracture network have been previously
characterised and interpreted from drone imagery. The studied outcrops are  considered as
analogues of the Potiguar Basin, Brazil (Bertotti et al., 2017; Bisdom, 2016). Uncertainties
were evaluated by comparing original outcrop interpretation (done manually by a geologist)
with the geometrical characteristics of the network generated from MPS. To evaluate the
quality of the simulations, we computed density maps in outcrop fracture interpretation and
on selected stochastic models. The proposed approach is innovative and provides a quick and
efficient way to represent fracture network arrangements at various scales.

**II] Methodology**
**II.1 The direct sampling method**
The direct sampling method (DS) was introduced by Mariethoz et al., (2010). Figure 1,
synthesizes the DS modelling process developed thereafter. The method requires a simulation
grid where each node is initially unknown and called $x$, a training image grid (TI) where each
node is known and called $y$ i.e. V(y) is defined where V is the variable of interest (e.g. facies
value). The simulation proceeds as follows. First, the set of conditioning data (if present) is
integrated in the simulation grid. Then, each remaining unknown node $x$ is visited following a
random or defined path, and simulated as follows. 1) The pattern $d_n(x) =$
$(x_1, V(x_1)),…,(x_n, V(x_n))$ formed by the at most **n** informed nodes the closest to $x$ is retrieved.
Any neighbour $x_i$ of $x$ is either a previously simulated node or comes from the conditioning
data set. The lag vectors $h_i = x_i\text{-}x$ define the geometry of the neighbourhood of $x$. The
combination of the value and position of $x_i$ defines the data event or pattern $d_n(x)$. 2) Then, the
TI is randomly scanned to search for a pattern $\mathbf{d_n(y)}$ similar to $\mathbf{d_n(x)}$. For each scan node $y$, the
pattern $d_n(y) = (y_1, V(y_1)),…,(y_n, V(y_n)),$ where $y_i=y+h_i,$ is compared to $\mathbf{d_n(x)}$ using a distance
(Meerschman et al., 2013). When the distance is lower than an acceptance threshold (**t**)
defined by the user or if the proportion of scanned nodes in the TI reaches a maximal fraction
(**f**) defined by the user, the scan is stopped and the value of the best candidate **y** (pattern with
the minimal distance) is directly attributed to **x** in the simulation grid (i.e. $V(x) = V(y)$).
As the DS method does not use a catalogue of all possible patterns found in the TI, it is
extremely flexible and in particular allows taking into account both categorical and
continuous variables and managing multivariate cases, provided that the pattern distance is
suitable. In this paper we are using the DeeSse version of the direct sampling code
(Straubhaar, 2017).

**II.2 Multiscale fracture attributes**
To evaluate how the direct sampling method deals with the fracture network, the present
experimentation is based on outcrop data where the present-day structural reality is
observable at various scales. Pavements (i.e. horizontal surfaces in the order of $10^2$ m scale)
were targeted because they contain important information that is not always accessible with
vertical outcrops (Corradetti et al., 2017a; Corradetti et al., 2017b; Tavani et al., 2016) or with
geophysical imagery (e.g. seismic data). The size of pavements allow the user to interpret a
large amount of fracture and to define areas where the geometry of the network varies (Bruna
et al., 2018). Pavements also allow to obtain quantitative data on fracture lengths, which are
usually difficult to get in vertical cliff. In the subsurface, data can be provided by geophysical
3D maps and fracture attribute detection tools (Chopra and Marfurt, 2007; Somasundaram et
al., 2017). However, these tools are not always available and detect the longer lineaments
only.
Working with pavements constitutes an asset as small-scale investigation can be conducted in
key zones of the outcrop (i.e. in folded areas, each compartment or dip domain of the fold
should be imaged and investigated in detail where the gathered data will help to calibrate
larger scale information. Classical fieldwork methods (observation and characterisation,
measurements, statistical analyses, sampling) help interpreting fracture families and are
essential to constrain larger scale observation.
In this study, UAV-based photogrammetry is used to obtain an orthorectified mosaic and 3D
digital outcrops models (Bemis et al., 2014; Claes et al., 2017; Vollgger and Cruden, 2016).
The scale of these images is an intermediate between the scale of measurement station and
that of satellite imagery. Digitization of fracture traces, geological contacts, sedimentary
structures and structural domain boundaries are currently processed by hand and represent a
considerable time investment. In this contribution, fractures were interpreted in orthomosaic
images with the help of GIS software. Length, azimuth, fracture family proportions and
fracture density statistics were extracted from the interpretation. In addition, a series of
measurement station (area of about $2 \times 2$ m) information was acquired and compared with the
dataset from the drone imagery in order to align interpretations and provide coherent fracture
history.

**II.3 Training images, conditioning data and probability maps**
• **Training images**
Training images (TI) are the base input data of the MPS simulation. Building them is a critical
step to succeed a realisation (Liu et al., 2009). The TI is a pixelated image based on a local
interpretation of a geological phenomenon (i.e. an interpreted photography taken from a local
zone of interest in the field) or digitised by a geologist and based on geological concepts
(Strebelle, 2002). As the MPS algorithms borrow patterns from the TIs to populate the
simulation grid, one should use TIs synthesising all of the recognized geological parameters
that characterise the area to simulate. To model non-stationary fields, i.e. fields where the
characteristics of the patterns differ depending on their location, one can follow two
strategies. The first one consists in using a non-stationary TI containing all wanted spatial
features. This requires to build one or several auxiliary variables describing the non-
stationarity in the TI and to define these auxiliary variables in the simulation grid to constrain
the simulation and indicate which kind of patterns will be simulated in which locations
(Chugunova and Hu, 2008; Mariethoz et al., 2010; Straubhaar et al., 2011). The second
approach consists in using several stationary TIs, each one depicting the same kind of patterns
everywhere, and defining zones in the simulation grid corresponding to each specific TI. This
second approach is chosen in this work, because it allows to define simple geological
concepts (TIs) specific to regions delineated in the simulation domain. The facies proportions
and their spatial arrangement belongs to each TI and can vary from one image to the other
(figures 5, 6, 9 and 10). Each TI has a local impact on the simulation. Moreover, in our
approach fractures sets are grouped in facies in the TI, based primarily on their orientation
and possibly on their length or additional parameters defined by the user. The fractures
classification helps reproducing patterns and simplifies the process of building the TIs. Note
also that two TIs used for two adjacent zones should share some common features in order to
obtain realistic transitions between the regions in the simulation domain.
• **Conditioning data**
One limitation of the MPS methods is the tendency to disconnect long continuous objects (i.e.
typically fractures, Bruna et al., 2017). To manage this issue, long fractures can be identified
and incorporated into the simulation as conditioning data. As per the training images, such
data can be integrated as pixelated grids. They may come from satellite imagery or they can
be interpreted from gravity or magnetic surveys or from 3D seismic imagery (Magistroni et
al., 2014).
• **Probability map**
The direct sampling method can be used with multiple training images. In this situation, the
user provides a set of TIs, and for each TI a probability map is defined on the simulation grid,

giving at each node the probability to use that TI. The pixel-wise sum of these maps should then be equal to one in every node. If each TI corresponds to a partition of the area of interest, with for each TI one elementary zone, covering the whole simulation grid, the probabilities in the map are set to one for specific TI and to zero for the other ones.

As per the training images, the probability map comes from a simple sketch (i.e. a pixelated image) given by the MPS user. It is based on the geological concepts or interpretations that define the geometry variability over the simulated area and that allow a partition of the outcrop. In each of the zones defined in the area of interest, the simulated property will follow the intrinsic stationarity hypothesis (Gringarten and Deutsch, 2001; Journel and Zhang, 2006; Journel, 2005) but the entire domain will be non-stationary.

While working on outcrops, the partition of the area of interest can be decided based on observations. For instance, when the fracture network interpreted from outcrop images is available, the geologist can visually define where the characteristics of the network are changing (fracture orientation, intensity, length, topology) and draw limits around zones where the network remains the same (internal variability, Hooker and Katz, 2015). However, in other cases outcrops or subsurface observation could be discontinuous between observation sites. If the data are sparse and come mainly from fieldwork ground observations or boreholes, the use of alternative statistical approaches can help to provide a robust and accurate partition of the area of interest. The work of Marrett et al., (2018) interprets the spatial organisation of fractures using advanced statistical techniques such as normalized correlation count and weighted correlations count, on scanlines collected in the Pennsylvanian Marble Falls Limestone. In their approach, the periodicity of fracture spacing (clustering) calculated from the mentioned techniques is evaluated using Monte Carlo to quantify how different the fracture networks are from a random organisation. These approaches can be highly valuable during the process of building a probability maps when less data are

available. The probability maps provide a large-scale framework that may be refined and
modified with additional data such as measurement stations or drone surveys coming from
surface exploration or wells data containing fracture network information.

**II.4 Testing the simulated network: from pixels to segments**
MPS realisations are produced as pixelated images. To evaluate the resulting fracture
network, pixels alignments corresponding to fractures are extracted as discrete straight-line
objects defined by start and end points. Fractures are separated from the background and in
different sets by automatic image classification methods. On grayscale images, this is
obtained by multilevel image thresholding through the Otsu's method (Otsu, 1979). On color
images, fracture sets are classified based on their color components with the k-means
clustering algorithm built in MATLAB (Lloyd, 1982). Image classification gives in output a
series of binary images, one for each fracture set, where lineaments are represented as
foreground (Kovesi, 2000).

**III] Results: test case on analogues of the Potiguar Basin, E Brazil**
**III.1 Geological setting**
The Potiguar Basin is a rift basin located in the easternmost part of the Equatorial Atlantic
continental margin, NE Brazil (fig. 2). The basin is found both onshore and offshore (fig. 2).
The basin was generated after the initiation of the South American and African breakup
during the Jurassic - Early Cretaceous times. It was structured by a first NW-SE extension
stage latterly rotating to an E-W extensional direction (Costa de Melo et al., 2016). The rift
basin displays an architecture of horsts and grabens striking NE-SW and bounded towards the
east and south by major fault systems (de Brito Neves et al., 1984), fig. 2). The Potiguar Basin
displays three sedimentary sequences deposited since the Early Cretaceous (i.e. syn- and post
rift depositions). The last post-rift sequence was deposited since the Albian and encompasses
the Cenomanian-Turonian Jandaíra Formation. This formation consists of up to 700 m thick
bioclastic calcarenites and calcilutites deposited in transgressive shallow marine environment.
From the Campanian to the Miocene, the compressive principal stress was oriented N-S
(Bertotti et al., 2017). From the Miocene to the Quaternary the onshore part of the Potiguar
basin was uplifted. Synchronously, a new compressive stress field was established trending to
a NW-SE direction (Reis et al., 2013).

**III.2 Outcrop data**
The area of interest measures $2.1 \times 1.3$ km and is located about 25 km NE of the city of Apodi
in the Rio Grande Do Norte state (fig. 2). It contains two outcrops AP3 and AP4 (Bertotti et
al., 2017; Bisdom, 2016, fig. 2) here defined respectively as $600 \times 300$ m and $400 \times 500$ m
large pavements localized in the Jandaíra Formation. AP3 and AP4 crop out as pavements
with no significant incision. The outcrops are sparsely covered by vegetation and
consequently they present a clear fracture network highlighted by karstification. In 2013,
images of AP3 and AP4 were acquired using a drone (Bisdom, 2016) and processed using the
photogrammetry method. Two high-resolution ortho-rectified images of these pavements
(centimetre-scale resolution) were used to complete fracture network interpretation and to
extract fracture parameters. In AP3, 775 lineaments were traced (fig. 3) and in AP4, 2593 (fig.
4). These lineaments collectively termed fractures in this paper. For each of these outcrops
three fractures sets were identified: set1 striking N135-N165, set2 striking N000-N010/N170-
N180 and set 3 striking N075-N105. Fractures falling outside of these ranges were not
considered in the input data. Consequently, in AP3 we considered 562 only (out of 775
fractures traced in the pavement) and in AP4 we considered 1810 only out of 2593 fractures.
In addition, ground-based fieldwork was conducted in AP3 and AP4 to understand the
structural history of the area and to calibrate the interpretation conducted on the drone aerial
photography. General location and fracture data are presented in figure 3 and 4 and in table 1.
In AP3, sets 1 and 2 are distributed over the pavement. However, their intensity is variable in
the area of interest. Set 3 is mainly expressed in distinct regions of the outcrop. Small-scale
investigations (conducted on measurement stations in the outcrop) showed that set 3 is
composed of stylolites and sets 1 and 2 of veins. In addition, sets 1 and 2 present evidences of
shear movements and are then considered as a conjugate system.
In AP4 small-scale investigations highlight the same characteristics as the ones observed in
AP3. Although the conjugate system (set 1 and set 2) is less developed there than in AP3. It is
also notable that more crosscutting relationships were observed in AP4 compared to AP3.

**III.3 Input data for MPS simulation**
To evaluate the effect of conditioning data, results of two simulations were compared, with
and without conditioning data. The sensitivity of simulation parameters was investigated by
varying i) the number of neighbours defining patterns (data events $d_n$), ii) the acceptance
threshold ($t$) defining the tolerance the algorithm authorises to find a matching data event in
the simulation grid (Mariethoz et al., 2010) and iii) the fraction of the TI to be scanned during
the simulation process to search for data events. Results of this sensitivity analysis help to
propose the best possible simulation for AP3 and to optimise the choice of input parameters
for AP4 fracture simulation.
AP3 presents intrinsic fracture network geometry variability. This observation emphasizes
that averaging fracture parameters on the entire domain is not well suited to represent the
complexity of the network. We observed that the length of fracture per sets and the density of
fractures are parameters that vary the most here. The analysis of these variations allow to
partition AP3 and AP4 in elementary zones and to synthesize the fracture network
characteristics in each of these domains. The following section defines how the TI,
probability map and conditioning data were built.
● **Partitioning, training images and probability map for AP3 and AP4**
We divided AP3 in 5 elementary zones (EZ) based on visual inspection of the pavement (fig.
5A-B). The number of fractures per EZ is synthesized in figure 5. The proportion of fracture
per elementary zone is available in table 1. A limited part of the fractures belongs to two
adjacent elementary zones. This issue is quantified in table 1.
A probability map with sharp boundaries (fig. 5B) was created for AP3. Sharp boundaries are
justified by the variability of the network geometry, which is known from the visual
inspection of the interpreted image. Smooth transitions could also be defined (see discussion).
The input data to build the probability map is an image of the partition of the area of interest
containing the different outcrops. In this image, the indexed zones (elementary zones EZ) are
characterised by a distinctive colour.
At the scale of a reservoir where some outcrops analogues and fracture tracing may be
available, the interpreted reality of the network (e.g. a binary fracture/non-fracture image) can
be directly used as a training image. We chose to ignore the tracing and to rely on parameters
that are attained through field observation without having access to drone images of an entire
outcrop (i.e. orientation, spacing, abutment) and to compare the interpretation with the
simulated network. In that respect fracture orientation were averaged to a single value. Hence,
set 1 strikes N090, set 2 strikes N150 and set 3 strikes N180. According to the outcrop
partitioning, five training images were created (fig. 5C). In each training image, three facies
corresponding to the three fracture sets were created. Set1 is green, set 2 is red and set 3 is
blue (fig. 5C). The topology is a crucial problem in fracture simulations because it influences
the connectivity of the network. In the MPS simulations the abutments are particularly well
reproduced as they represent singular pixels arrangements that are efficiently taken into
account. However, crosscutting relationships imply the use of a different facies at the
intersection locus. This method respects and reproduces intersections during the simulation
process. In AP3, the analysis of the topology relationships showed three main crosscutting
interactions:
- Long fractures from Set 2 and Long fractures from Set 3 mutually crosscut
(conjugated sets)
- Set 3 crosscut Set 1
- Set 2 crosscut Set 1
To take into account these topological parameters a different facies colour was attributed to
the crosscutting locus (the crossing facies, fig. 6). When the MPS realization will be later
discretized, the younger fractures will be truly represented as continuous segments. The older
fractures will be cut in pieces but their alignment will be, in most of the case, maintained
during the simulation process.
• **Dimensions of the simulation grids and of the training images**
The dimensions of the simulation grid for AP3 and of each training image (in pixels) are
shown in fig.5. The number of pixels is automatically determined by the size of the original
drawing made by the geologist.
The size of the input training image does not generally influence the simulation. However, it
has to be chosen sufficiently large with respect to the complexity of the patterns in order to
get reliable spatial statistics. The DS method tends to identify patterns (i.e. $d_n$'s see above) in
the TI and to paste the central node of them into the simulation grid. However, at a constant
resolution and specifically for fractures patterns, it is likely that a $50 \times 50$ m training image
will carry more complexity and variability than a $10 \times 10$ m one. This parameter should be
taken into consideration when starting digitizing training images, especially when spacing
between fractures is not consistent across the simulation grid.
● **Long fractures conditioning**
Because the MPS method has the tendency to cut long individual segments into smaller
pieces, the fractures longer than 40 meters – the ones visible from satellite/drone imagery in
AP3 – were isolated and considered as hard conditioning data (fig. 5D). This threshold was
arbitrarily determined from the dataset we have. In AP3, less than 8% of the fractures are
longer than 40 m.
In AP3, long fractures belong only to the sets oriented/striking N180 or N150 (fig. 5D). 18
N180 fractures (3% of the whole) and 30 N150 fractures (5% of the whole) were digitized and
integrated as conditioning data in the simulation.

**III.4 Outcrop scale simulations**
**III.4.1 Impact of conditioning data on AP3 simulations**
In AP3, the 48 long fractures were manually digitized and imported into the simulation grid as
categorical properties to be considered as hard conditioning data during the MPS simulation
process. The MPS simulation is consequently in charge of stochastically populating the
smaller factures within the grid.
Results of the influence of these data are presented in figure 7. The principal simulation
parameters in the considered scenarios (with and without conditioning data) were set up
identical (constant acceptance threshold (5%), constant percentage of scanned TI (25%) and
constant number of neighbours (50)).
Results showed that the realisation without conditioning data creates 20% less number of
fractures than the original outcrop reference. The simulation with conditioning data creates
9% less number of fractures than AP3, which allow to better replicate the long fracture than a
non-conditioned simulation. It is also remarkable that the non-constrained simulation
represents only 23 fractures above 40 meters (compared to the 48 long fractures interpreted on
the AP3 outcrop). In this simulation the long fractures are essentially located in the zone 3 of
the outcrop. Because the simulation is a stochastic process, the location of the long fractures
is randomly determined in the absence of hard conditioning data. Considering hard-
conditioning data also gives a more realistic representation of the fracture network.

### III.4.2 Sensitivity analysis on the AP3 simulation parameters

• **Simulation parameter set-ups, duration and analyses conducted on the results**
Simulation parameters were varied for each simulation in order to emphasize their effect on
each realisation. One realisation per test was performed during this analysis. The goal of this
analysis is to show how the different parameters influence the reproduction of fracture
segments and not to evaluate how good is the matching between the simulation and the
reference.
The MPS realisations are pixelated images. The sensitivity analysis is based on the discrete
segments extracted from these pixelated images (see II.4).  All of the simulations present a
variable percentage of segment lengths that are below the minimal fracture length interpreted
in the AP3 outcrop (i.e. simulation noise). Consequently all segments smaller than 2.2m
where removed from the simulation results. A length frequency distribution was compiled for
each of the generated simulations.
The influence of the number of neighbours was evaluated trough 7 simulations (SIM1 to
SIM7). The acceptance threshold and the number of neighbours was investigated by
comparing 8 simulations (SIM8 to SIM15) where the scanned fraction of the TI was fixed at
25%. The percentage of the scanned fraction of the TI was combined with the two other
simulation parameters. This combination was tested over 12 simulations (SIM16 to SIM27).
The models set-ups and the duration of the simulations are presented in (table 2). It is notable
that SIM8 / SIM9, SIM10 / SIM11 and SIM13 / SIM14 produce exactly the same network
despite the modification of the simulation parameters. Also The MPS algorithm successfully
performed SIM16 but the segment extraction generated an error preventing the discretisation
of all of the objects.
The total amount of generated fractures segments was counted and compared with the total
amount of fracture traces interpreted from the original outcrop. A deviation of 10% compared
to the original amount of interpreted fractures is considered as a satisfactory result as it is very
close to the reference amount of fractures. A deviation of 20% compared to the original
amount of interpreted fractures is considered as an acceptable result. This deviation is
consequent but can be adjusted by varying the simulation parameters. A deviation above 20%
was rejected as a complete reconsideration of the parameters is required. Results are
synthesized in table 3.
The total amount of segments was initially counted in the entire simulation domain. The sum
of segments per part is constantly higher than the initial total amount of segments because
segments cutting a sharp boundary are divided in two - segments falling within two
elementary zones and are consequently counted twice. The number of generated fractures per
simulation zone was also computed and the same deviation thresholds were applied to
evaluate if the simulation is satisfactory, acceptable or rejected. Tables 4 to 6 synthesize the
sensitivity analysis conducted of 27 realisations of the AP3 outcrop.
The length of the segments have been computed for each realisation and are presented in
figure 8.
The influence of the hard conditioning data and of the drawing of the training image was also
quantitatively investigated and compared respectively with the length of the generated
segments and with the amount of segments generated per zone.
• **Summary of the results**
Increasing the number of neighbours lengthens the computation time (table 2, SIM 1 to 7). A
small amount of neighbours results in a noisy simulation (table 2, SIM1). The contrary leads
to a downsampling of the generated segments that become longer than the interpreted
fractures in AP3 (table 2, SIM7). Decreasing the acceptance threshold leads to an increase of
the simulation time (table 2 SIM8-15). Increasing the scanned fraction of the TI is the most
time consuming operation (table 2 SIM17-27).
Increasing the number of neighbours only is generally not sufficient to accurately generate a
satisfactory or acceptable total amount of fractures (table 3). Increasing the scanned fraction
of the TI produces in all cases the closest total number of fractures compared to the reference
outcrop (table 3).
The counting of fractures in simulation zones revealed that set 2 and set 3 in zone 1, set 3 in
zone 4 and set 1 in zone 5 are generally underestimated during the simulation process. In
contrast, fracture set 1 in zone 2 is generally overestimated. The consistency of the error over
almost the entire set of simulations indicates an issue on the training image representation
(table 4-6). Increasing the scanned fraction of the TI generally allows to better represent a low
proportion of fracture facies within a TI (Zone TI5, set 2, table 6).
An acceptance threshold below 5% leads to an overestimation of the number of small
fractures (between 0-10 m), fig 8. In this case, amount of segments between 0-20 m is
generally close to the reality. Increasing the scanned fraction of the TI produces the highest
quantity of fractures ranging from 0-10 m (fig. 8). Increasing the number of neighbours and
the percentage of the scanned TI will result in an increase of the length of the fractures used
as hard conditioning data. However, the fracture elongation does not affect all of the hard
conditioned fractures and represents a very small percentage of the whole modelled fracture
network.

### III.4.3 Attempt at an optimisation: OPT1


OPT1 was parameterised in regard of the previous observations in order to generate a
simulation that is the closest-to-reality possible. For this purpose, the amount of fractures
from set 2 and set 3 drawn in TI1 and set 3 drawn in TI4 was increased. In contrast, the
amount of fractures from set 1 drawn in TI2 was decreased significantly (fig. 9). We choose
to setup the number of neighbours at 50 and the acceptance threshold at 2%. TI1 and TI4 will
be scanned at 75% and the rest of the TIs will be scanned at 50% (table 2).
The simulation time for the proposed simulation is 2 min 31s (table 2). The total amount of
generated fractures is satisfactory compared to the amount of fractures interpreted in the
original outcrop.
To evaluate the robustness of the optimised simulation, 6 realisations using the same
parametrisation were generated for OPT1. The total amount of fractures generated for these
simulations always fall below the 10% deviation compared to the reference outcrop.
The number of segments comprised between 0-20 m in OPT1 is slightly above the
satisfactory deviation limit. As per all the generated simulations, the number of fractures
between 2.21 m and 10 m is largely overestimated.
OPT1 contains a more satisfactory and acceptable fracture count than any other simulation
generated before (table 6). The amount of segments generated in zone 1 and 2 for set 1 is
slightly overestimated. In zone 3, OPT1 fails to represent the amount of fractures for set 1
(25% deviation) and for set 3. Fracture set 1 in zone 4 is largely overestimated.

### III.4.4 Evaluation of the AP3 and OPT1 simulations: $P_{21}$ calculations


Uncertainty analysis is required when performing simulations of geological parameters,
especially far from data. The sensitivity analysis presented in this paper is a way to compare
the MPS simulations with the reference outcrop.

To reinforce the evaluation of the proposed method, we quantified the values of fracture intensity in the reference outcrop, in three selected AP3 MPS simulations and in the optimised simulation (OPT1) (fig. 10). The fracture intensity was classified by (Dershowitz and Herda, 1992) in regard of i) the size and dimension (1D, 2D, 3D) of a selected zone of interest and ii) the number, length, area or volume of fractures within this selected zone. In this paper, we chose to calculate the $P_{21}$ fracture intensity, which corresponds to the sum of all fracture lengths within a regularly discretized space, with constant area boxes ($10 \times 10$ m) covering the entire AP3 area of interest.

Visually, the results show an apparent higher $P_{21}$ intensity in the reference outcrop than in the simulations. However, zones of high intensity in the reference outcrop are generally well represented in SIM26 and in OPT1. This is in agreement with the results of the sensitivity analysis showing that SIM26 and OPT1 best represent the number of fractures present in the reference outcrop.

The average fracture intensity in each simulation has also been computed and confirms the observations conducted during the sensitivity analysis. SIM1 and SIM7 present the lowest average fracture intensity (0.095 m$^{-1}$ and 0.079 m$^{-1}$ respectively) and SIM26 and OPT1 present the highest fracture intensity (0.11 m$^{-1}$ and 0.099 m$^{-1}$ respectively). The average fracture intensity in the reference outcrop is higher than in any other simulations (0.126 m$^{-1}$). However, this value remains close to the ones obtained in SIM26 and OPT1.

The fact that the fractures have been simplified as straight lines in the simulations combined to a relatively small area of calculation ($10 \times 10$ m) could be one element of explanation of the observed fracture intensity variation between the reference outcrop and SIM26 and OPT1. This analysis strengthens the results obtained during the sensitivity analysis and demonstrates the capacity of the MPS method to represent with a high fidelity the geometry of a fracture network.


### III.4.5 Using the sensitivity analysis results to model AP4

As per AP3, AP4 present an intrinsic variability of the fracture network geometry. This
outcrop was divided in 3 elementary zones (fig. 11A-B). According to AP4 partitioning, a
probability map with sharp boundaries (fig. 11B) was created. For AP4, the configuration of
the outcrop led to mask the area where no interpretation data were performed. In these
particular zones a "no data value" was attributed and these masked areas were excluded
during the modelling process. In AP4 three training images were created (fig. 11C). As per
AP3, the size of the AP4 simulation grid was doubled compared to its original dimension
(available in fig.11). In AP4, fractures longer than 40 meters were also considered as hard
conditioning data. Here, less than 1.5% of the fractures are longer than 40m (fig. 11D). In
AP4, long fractures were found in the 3 sets and mainly in the south-eastern part of the
outcrop (fig. 11D, elementary zone 6). 11 N180 fractures (0.5% of the whole), 13 N150
fractures (0.6% of the whole) and 9 N090 fractures (0.4% of the whole) were digitized and
integrated as conditioning data into the simulation.
Based on the results of the sensitivity analysis of AP3 we generated one simulation for the
AP4 outcrop (fig. 12). The modelling parameters for SIM AP4-1 were selected as following:
the number of neighbours was set up at 50 and the acceptance threshold at 2%. The 3 training
images used in the simulation are presented in figure 12 and are considered as representative
of the fracture arrangement in each region of the simulation. The scanning percentage of TI6
and TI7 was set up at 50%. The scanning percentage of TI8 was set up at 100%. With this
configuration, the simulation lasts slightly more than 5 minutes. The fact of intensely
scanning TI8 is probably responsible of this duration. The analysis was conducted on the total
amount of segments generated and of segments per set of fractures. In AP4 the total number
of segments is 1810. The simulation realises 1682 segments in total, which constitutes a
satisfactory result. The original AP4 presents 252 segments striking N150, 856 segments
striking N180 and 702 segments striking N090. The results of simulation AP4-1 are always
satisfactory or acceptable with 206 segments striking N150, 834 segments striking N180 and
642 segments striking N090. A detailed analysis was not conducted here because AP4
contains a lot of small fracture intersections (especially in the TI8 zone) and this makes the
segment extraction a complex process. However, these results are promising for the future.

**IV] Smooth transitions between elementary zones: towards reservoir scale**
**models to manage uncertainties**
The strength of the method proposed here relies on the use of a probability maps and on the
opportunity to consider multiple training images in a single realisation to generate non-
stationary models of fracture network geometries. In the case of AP3 and AP4, the probability
maps are essentially constrained by the variation of geometry of the fracture networks
observed on the geological interpretation made on the drone imagery. Consequently, the
defined areas are pragmatically bounded and the nature of the limit between one zone and
another is a sharp boundary.
AP3 and AP4 outcrops are separated by about 2.5 km and very little is known about the
fracture network geometry between these two locations. Assuming that there is no major
structural deformation (fold or faults) that may cause a change in fracture geometry at the
close vicinity of the outcrop "reality", the zones initially defined on the AP3 and AP4 outcrop
can be extended to the limits of the reservoir-scale model boundaries (fig. 13). In this
particular case, filling the gap between the two outcrops appears to define how the transition
between one side of the simulation grid and the other should be determined.
Fractures are localised objects that do not need to be necessarily continuous from one
simulation zone to another. The constant higher proportion of the non-fractured matrix facies
versus localised and thin fracture elements ensures the coherency and relative compatibility
from one simulation region to another. The idea of the simulation grid region partitioning was
re-evaluated and an alternative method, was proposed here. Contrarily to the definition of
sharp boundaries in the probability maps used for AP3 and AP4, a probability map with
smooth transitions is defined as follows. An ensemble of elementary zones covering a part of
the simulation grid is defined. Each TI corresponds to one elementary zone, which is
simulated using exclusively that TI. The probabilities in these zones are then set to one for a
specific TI and to zero for the other TIs. The remaining part of the simulation grid is divided
in transition zones, for which one has to define which TIs may be involved. In a transition
zone, the probabilities of the involved TIs are set proportional to the inverse distance to the
corresponding elementary zones. This process creates smooth transitions in low constrained
area decreasing the influence of one TI towards another (from one elementary zone to
another).
No faults or folds can be initially identified between AP3 and AP4 to condition the drawing of
the probability map. In this case, a rectangular compartment representing a gradual
probability transition to use the training image associated to one outcrop or to the other filled
the blank space between the two outcrops. For instance, fig 13E shows in the
Transition_Zone_1 a decreasing probability to use TI1 from left to right (i.e. zone 1 to zone 6)
and conversely to use TI6 from right to left.
Recently, investigations conducted on the Rio Grande do Norte geological map (Angelim et
al., 2006), demonstrated the presence of a fault crossing the simulation grid near the AP3
zone. This structure may explain the variability of fracture geometry from AP3 (EW stylolites
and strong presence of conjugated NS/NW-SE system) to AP4 (EW stylolites associated to
NS fracture system, the NW-SE conjugated system is here subordinate). Further geological
investigations need to be conducted in this particular place to proof the influence of this fault
on the network geometry. However, fig 13F shows an alternative probability map taking into
account this interpretation and present how flexible the probability map can be. The proposed
method demonstrates its adaptability in various geological contexts.

**V] A method to create a 3D DFN out of 2D MPS realisations**
The MPS simulations presented in this paper are on the form of 2D pixelated maps.
MATLAB codes were developed to extract starting and end point coordinates (georeferenced)
of a series of aligned colorized pixels that represent a fracture trace from these images.
Transforming this output in geologically realistic 3D surfaces is not easy. Karimpouli et al.,
(2017) studied samples coming from coalbed methane reservoirs in the fractured Late
Permian Bowen Basin in Australia. They realised multiple 2D and pseudo 3D images (i.e.
orthogonal 2D images) and used the cross-correlation based simulation (CCSIM) to represent
the internal organisation of coal cleats and the heterogeneity of the coal matrix in 3D. Their
approach greatly improved the understanding of the internal complexity of coal samples and
gives better results than classical DFN's based on averaged distributions. However, their
method requires an important initial amount of information (i.e. CT scans slices used as
training images) that is generally not available at a larger scale. The use of MPS in 3D seems
particularly not suited for fracture network representation because: i) they require to associate
fractures from 2D map view and from 2D section view (3D or pseudo-3D), ii) it appears
difficult to consider isolated fractures in this type of approach and iii) in the subsurface
fracture height and/or fracture length are generally unknown.
To Tackle these problems we choose to use multiple 2D MPS-generated fracture networks. In
the presented approach, the 3D is obtained by extruding 3D fracture planes in fracture units
(fig. 14). In this approach we consider that fractures are entirely bound to the units, which can
appear as a limitation if isolated fractures occurs inside a layer. However, we can consider
variable levels of fracture units. Figure 14 presents an hypothetic scenario where red fractures
are confined to a large fracture unit (FU1) crosscutting smaller ones (FU4 containing also
smaller red fractures). In such a representation, one 2D planar simulation is required at each
top mechanical unit to generate a new set of fractures.
In real-world subsurface configurations, mechanical units can be extracted from well logs
(resistivity, density, lithology; Laubach et al., 2009). The fracture height distribution, referred
as fracture stratigraphy (Hooker et al., 2013) requires here a particular attention and is
difficult to extract from borehole data. In outcrops, the use of vertical cliffs adjacent to 2D
horizontal pavement should be a way to evaluate these heights and to constrain the 3D model.
In outcrops, the resort to vertical cliffs adjacent to 2D horizontal pavements is required to
define fracture height. This method is already implemented in gOcad-SKUA software as a
macro that extrudes planes of a single fracture family (i.e. all the red fractures in AP3)
vertically into a bounded volume (fig. 14). More developments are in process to generate
oblique planes and to be able to extrude planes in portions of the fracture sets.

**V] Conclusions**
In this paper a new method to predict the geometry of a natural fracture network using the
multiple-point statistic algorithm is presented. The method provides stochastic realisation
depicting a realistic non-stationary fracture network arrangement in 2D based on the use of
multiple, simplified, small training images capturing the natural fracture attributes in specific
zones defined by a probability map. Probability maps are adaptable and follow geological
rules of fracture type and arrangement distribution specific to various tectonic contexts (i.e.
faulting, folding and poor deformation context/no fault, no folds). We developed methods to
be able to consider transition zones into the probability maps (e.g. zones far from hard data)
that allow simulating fracture network geometry at a larger scale (i.e. reservoir scale).

The realisations obtained from 2D MPS constitute a statistical laboratory close enough to the reality to be tested in terms of fracture mechanical parameters and response to flow. Comparison between mechanical aperture calculation, fluid flow simulations conducted on both "reality" fracture network interpretations performed on drone imagery and series of MPS realisations gives similar results.

The method proposed here is applicable to all rock types and to a wide range of tectonic contexts. Initially calibrated using outcrop data, the method is fully adaptable to the subsurface in order to better characterise fractures in water, heat or hydrocarbon reservoirs. The challenge there, remains on the definition of the different training images on which the simulation is based. Very few data are generally available in the subsurface and geological rules need to be found to define the geological characteristics of the fracture network (orthogonal or conjugate network) and the associated fracture attributes (length, height, spacing, density, topology).

## Acknowledgments

The authors want to thank ENI S.P.A. for the financial support of this research. Silvia Mittempergher from the University of Milano Bicocca is acknowledged for providing the code extracting segments from pixelated images. We would like also to thank the entire SEFRAC group for their interest in developing this method and for their valuable geological advices. Acknowledgements are extended to Philippe Renard from the University of Neuchâtel, to Hadi Hajibeygi from TU Delft and to Wilfried Tsoblefack from Paradigm Geo for the constructive discussions we had together. Prof. Hilario Bezerra from the Universidade Federal do Rio Grande do Norte is acknowledged for providing datasets concerning Apodi area and for his advises on the local geology. We would like to thank Jan Kees Blom from the TU Delft for the improvement he provided to this manuscript. We thank the two anonymous

reviewers, Stephen Laubach, William Dershowitz and John Hooker for their very useful
comments that greatly participated to improve this paper.

**Appendix A**
The DeeSse algorithm (Straubhaar et al., 2011) was used in this paper to reproduce existing
fracture network interpreted from outcrop pavements. The following pseudocode  developed
by Oriani et al., (2017) have been modified to explain how the algorithm is processing the
simulation of fracture. Specific terms can be found in section II.1 of the present paper. In our
study the simulation follows a random path into the simulation grid. This grid is step by step
populated by values (fracture facies in our case) sampled in the training image. The algorithm
proceeds according to the following sequence :

1. Selection of a random location $x$ in the simulation grid that has not yet been simulated

(and not corresponding to conditioning data points, already inserted in the grid).

2. To simulate $V(x)$ → the fracture facies into the simulation grid: The pattern $d_n(x) =$

$(x_1, V(x_1)), ..., (x_n, V(x_n))$ formed by at most $n$ informed nodes the closest to $x$ is retrieved.

If no neighbours is assigned (at the beginning of the simulation), $d_n(x)$ will then be empty:

in this case, assign the value $V(y)$ of a random location y in the TI to $V(x)$, and repeat the

procedure from the beginning.

3. Visit a random location y in the TI and retrieve the corresponding data event $d_n(y)$.

4. Compare $dn(x)$ to $dn(y)$ using a distance $D(dn(x), dn(y))$ corresponding to a measure of

dissimilarity between the two data events.

5. If $D(dn(x), dn(y))$ is smaller than a user-defined acceptance threshold $T$, the value of

$V(y)$ is assigned to $V(x)$. Otherwise step 3 to step 5 are repeated until the value is assigned

or an given fraction $F$ of the TI, is scanned.

6. if **F** is scanned, **V(x)** is defined as **V(y)**, with **y** the scanned location minimising the
distance **D(*dn(x), dn(y)*)**.
7. Repeat the whole procedure until all the simulation grid is informed.

## Figure captions

**Figure 1:** Direct Sampling method workflow applied to fracture network modelling (modified
from Meerschman et al., 2013).

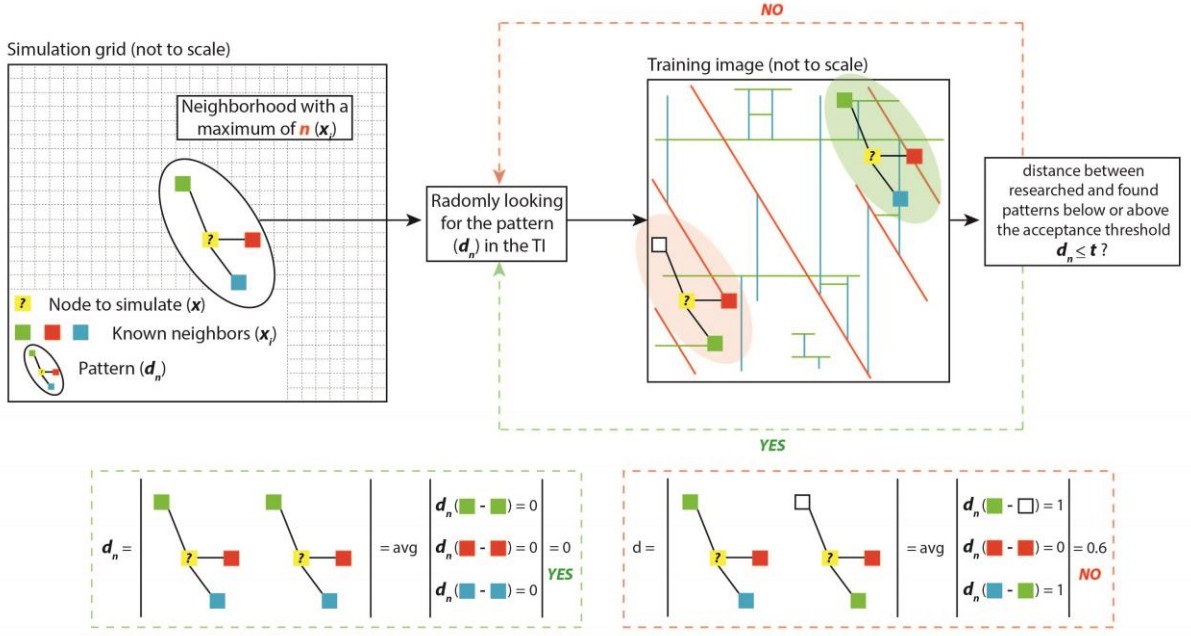



**Figure 2:** Location of the area of interest and of the studied pavements near Apodi area (red
star).

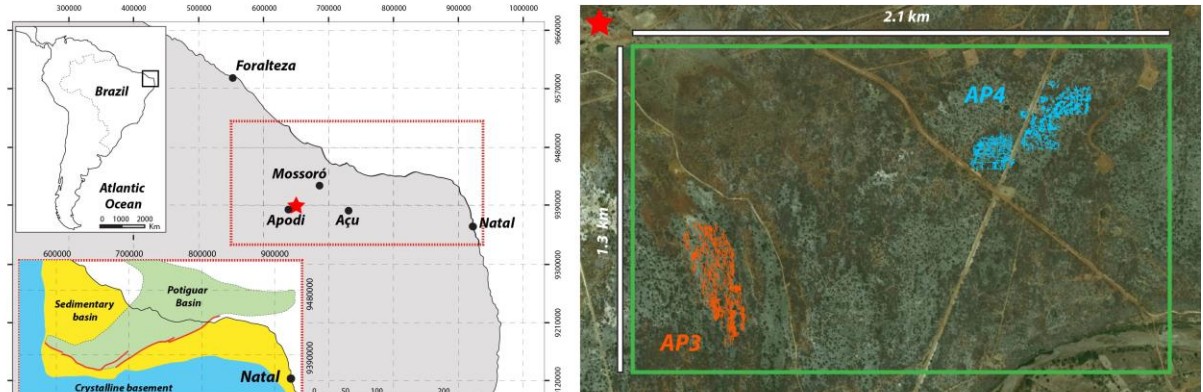






**Table 1:** Outcrop characteristics and fracture parameters collected in AP3 and AP4

| AP3 outcrop | | | | | | | | | |
|---|---|---|---|---|---|---|---|---|---|
| Localisation (WGS84 UTM Z24S) | | Orientation | Dimension | | Fractures proportion (of the whole fracture population) | | | | Fracture length |
| X | Y | | NS (m) | EW (m) | Set 1 (N135-N165) | Set 2 (N000-N010/N170-180) | Set3 (N075-N105) | Min (m) | Max (m) |
| 650601 | 9387908 | NNW-SSE | 600 | 300 | 30% | 52% | 18% | 2,21 | 123 |

| Elementary zone 1 | Elementary zone 2 | Elementary zone 3 | Elementary zone 4 | Elementary zone 5 | Elementary zone 1 | Elementary zone 2 | Elementary zone 3 | Elementary zone 4 | Elementary zone 5 | Elementary zone 1 | Elementary zone 2 | Elementary zone 3 | Elementary zone 4 | Elementary zone 5 |
|---|---|---|---|---|---|---|---|---|---|---|---|---|---|---|
| 60% | 26% | 18% | 70% | 87% | 37% | 14% | 80% | 23% | 13% | 3% | 60% | 2% | 7% | 0% |

| AP4 outcrop | | | | | | | | | |
|---|---|---|---|---|---|---|---|---|---|
| Localisation (WGS84 UTM Z24S) | | Orientation | Dimension | | Fractures proportion (of the whole fracture population) | | | Fracture length | |
| X | Y | | NS (m) | EW (m) | Set 1 (N135-N165) | Set 2 (N000-N010/N170-180) | Set3 (N075-N105) | Min (m) | Max (m) |
| 652032 | 9388508 | NE-Sw | 400 | 500 | 20% | 40% | 40% | 1 | 186 |

| Elementary zone 6 | Elementary zone 7 | Elementary zone 8 | Elementary zone 6 | Elementary zone 7 | Elementary zone 8 | Elementary zone 6 | Elementary zone 7 | Elementary zone 8 |
|---|---|---|---|---|---|---|---|---|
| 8% | 20% | 10% | 43% | 45% | 53% | 49% | 35% | 37% |

**Figure 3:** Data acquired in the area of interest in pavements AP3. A) ortho-rectified high-resolution pavement aerial images acquired with a
drone, B) fracture interpretation on ortho-rectified images, C) fracture orientation calculated from the north in GIS-based environment.
Corresponding rose diagram for both outcrops, D) length of each fracture trace and E) fracture topology relationship for each pavement observed
on fracture network interpretation.

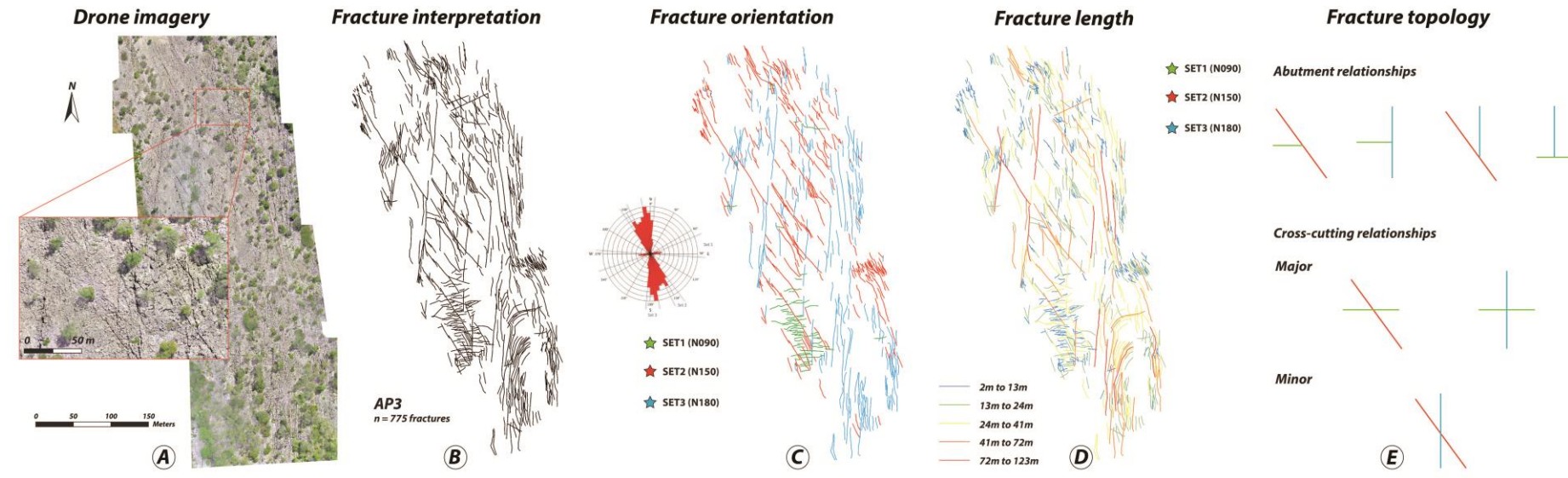





**Figure 4:** Data acquired in the area of interest in pavements AP4. F) ortho-rectified high-resolution pavement aerial images acquired with a
drone, G) fracture interpretation on ortho-rectified images, H) fracture orientation calculated from the north in GIS-based environment.
Corresponding rose diagram for both outcrops, I) length of each fracture trace and J) fracture topology relationship for each pavement observed
on fracture network interpretation

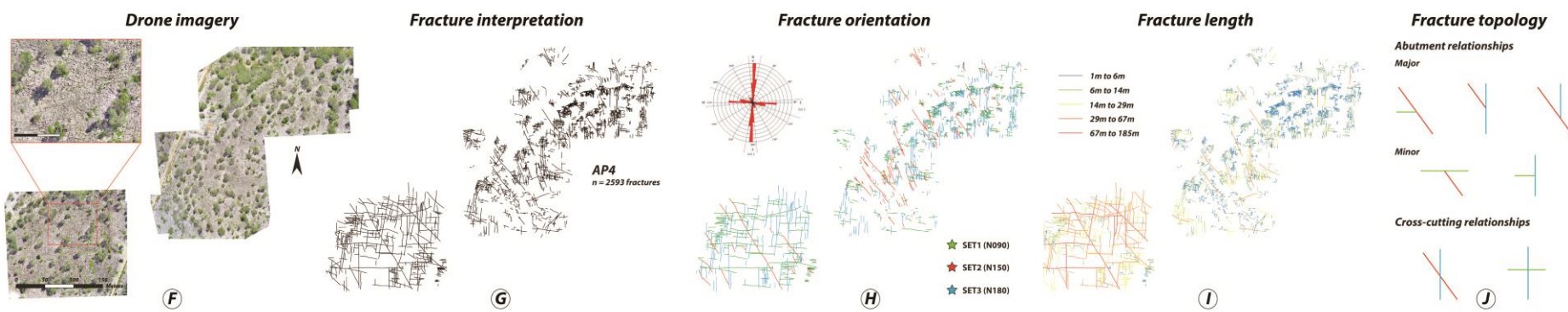







**Figure 5:** A) Partitioning of AP3 in 5 elementary zones (EZ). This partition is defined (with
respect to fracture orientation (fracture facies), fracture density and geometry variability over
the entire simulation domain. B) probability map and associated statistics for each EZ. C)
training images associated with the partition of AP3. In each EZ, the corresponding training
image has a probability (pTI) of 1 to be used. In this zone the other training images are not
used (pTI = 0). D) hard conditioning data for AP3. All the fractures longer than 40 m are
considered deterministically in the simulation process

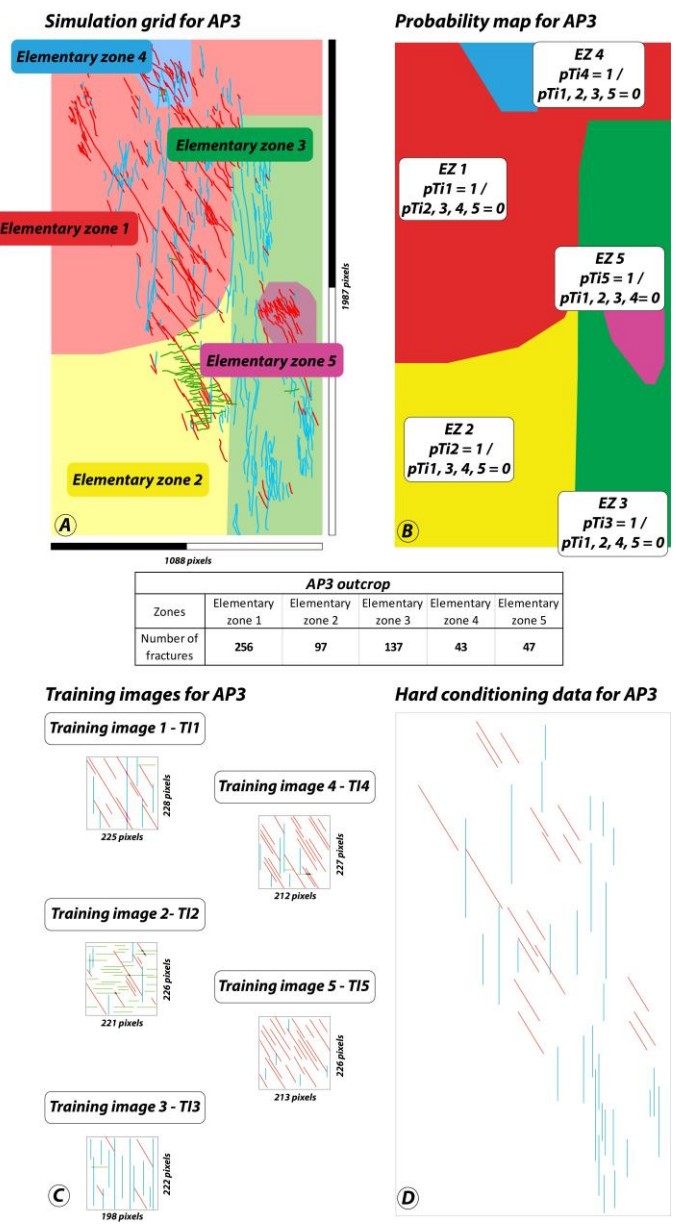


**Figure 6:** Comparison between results obtained without constraining the topology and with
topological facies constraints.

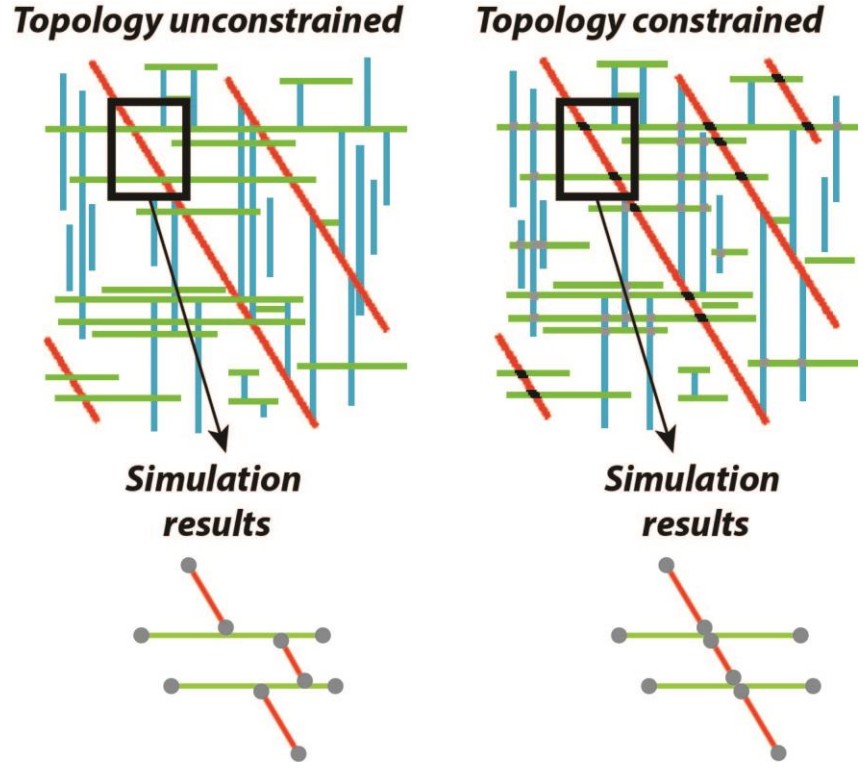



**Figure 7:** Visual comparison between: A) the reference fracture network interpretation (AP3),
B) the extraction of the longer segments (50 fracture longer than 40m), C) a simulation
conditioned by the long segments, D) a simulation not conditioned by the long segments

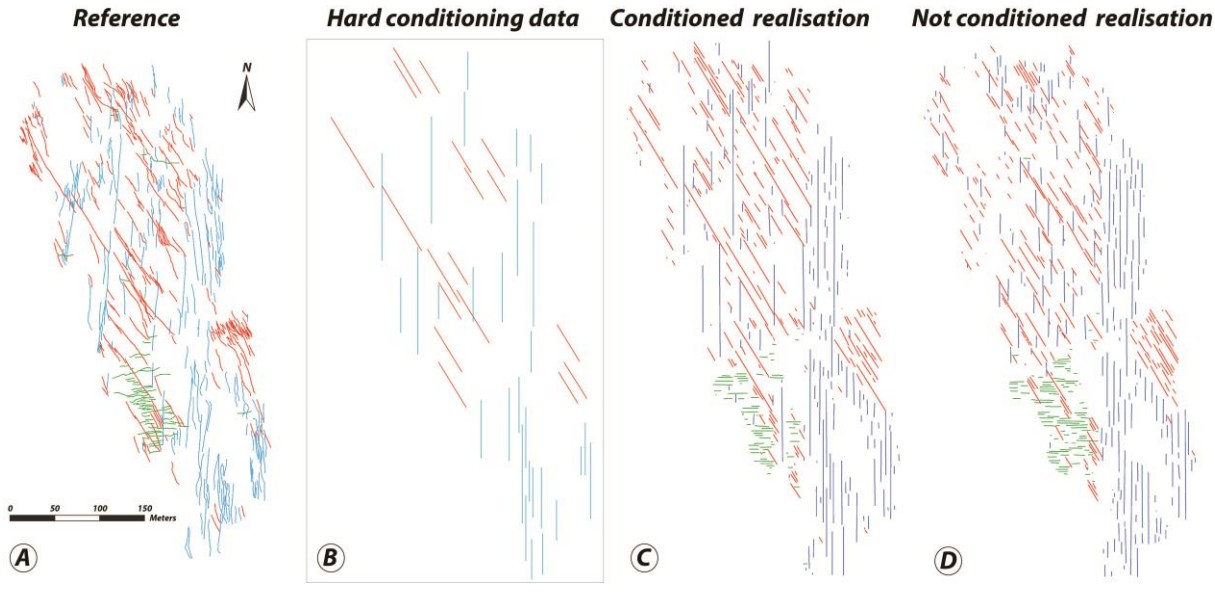


Table 2: Simulation parametrisation, models set-ups and duration (in seconds) of each run.

| Tested parametrisation | Number of neighbours influence | | | | | | | Number of neighbours + Acceptance threshold | | | | | | | |
|---|---|---|---|---|---|---|---|---|---|---|---|---|---|---|---|
| Realisation name | SIM1 | SIM2 | SIM3 | SIM4 | SIM5 | SIM6 | SIM7 | SIM8 | SIM9 | SIM10 | SIM11 | SIM12 | SIM13 | SIM14 | SIM15 |
| Simulation parameters | A. th. = 5% N. = 10 Scan= 25% | A. th. = 5% N. = 20 Scan= 25% | A. th. = 5% N. = 30 Scan= 25% | A. th. = 5% N. = 40 Scan= 25% | A. th. = 5% N. = 50 Scan= 25% | A. th. = 5% N. = 75 Scan= 25% | A. th. = 5% N. = 100 Scan= 25% | A. th. = 4% N. = 40 Scan= 25% | A. th. = 3% N. = 40 Scan= 25% | A. th. = 2% N. = 40 Scan= 25% | A. th. = 1% N. = 40 Scan= 25% | A. th. = 4% N. = 50 Scan= 25% | A. th. = 3% N. = 50 Scan= 25% | A. th. = 2% N. = 50 Scan= 25% | A. th. = 1% N. = 50 Scan= 25% |
| Simulation duration | 22" | 19" | 33" | 36" | 55" | 101" | 136" | 52" | 52" | 90" | 95" | 56" | 76" | 76" | 121" |

| Tested parametrisation | Number of neighbours + Acceptance threshold + % TI scan | | | | | | | | | | | | Optimisation |
|---|---|---|---|---|---|---|---|---|---|---|---|---|---|
| Group | Group 1 | | | | Group 2 | | | | Group3 | | | | |
| Realisation name | SIM16 | SIM17 | SIM18 | SIM19 | SIM20 | SIM21 | SIM22 | SIM23 | SIM24 | SIM25 | SIM26 | SIM27 | OPT1 |
| Simulation parameters | A. th. = 3% N. = 40 Scan= 50% | A. th. = 2% N. = 40 Scan= 50% | A. th. = 3% N. = 50 Scan= 50% | A. th. = 2% N. = 50 Scan= 50% | A. th. = 3% N. = 40 Scan= 75% | A. th. = 2% N. = 40 Scan= 75% | A. th. = 3% N. = 50 Scan= 75% | A. th. = 2% N. = 50 Scan= 75% | A. th. = 3% N. = 40 Scan= 100% | A. th. = 2% N. = 40 Scan= 100% | A. th. = 3% N. = 50 Scan= 100% | A. th. = 2% N. = 50 Scan= 100% | Custom |
| Simulation duration | 80" | 148" | 123" | 124" | 105" | 196" | 152" | 154" | 104" | 203" | 150" | 149" | 151" |

Table 3: Comparison between the total amount of segments interpreted in the reference outcrop and in the different sets of simulations (tested parametrisation). Evaluation of the results in terms of satisfactory (green symbol), acceptable (orange symbol) or non-satisfactory (red symbol)

| | Reference outcrop | Tested Parametrisation | Number of tested configurations | Results evaluation | | |
|---|---|---|---|---|---|---|
| | | | | ✔ | ≈ | ✘ |
| Total segments | 562 | Influence of the number of neighbours | n=7 | 1 | 1 | 5 |
| | | Number of neighbours + Acceptance threshold | n=8 | 3 | 2 | 3 |
| | | Number of neighbours + Acceptance threshold + % TI scan | n=12 | 5 | 6 | 1 |

**Table 4**: Results of the sensitivity analysis on the influence of the number of neighbours. The
table presents the number of segments per simulation zone for AP3 (used as reference). Red
symbols show a total amount of segments of the considered set in the considered zone
deviating to more than 20% from the reference case. Yellow symbols show a deviation of
more than 10% from the reference case. Green symbols do not deviate significantly from the
reference outcrop interpretation.

| | | | Number of neighbours | | | | | | |
|---|---|---|---|---|---|---|---|---|---|
| | | Reference | SIM1 | SIM2 | SIM3 | SIM4 | SIM5 | SIM6 | SIM7 |
| | | Segments per parts | | | | | | | |
| Zone TI1 | Set1 | 156 | ✗ | ≈ | ≈ | ✗ | ✗ | ✗ | ✗ |
| | Set2 | 95 | ✗ | ✗ | ≈ | ✗ | ✗ | ✗ | ✗ |
| | Set3 | 6 | ✗ | ✗ | ✗ | ✗ | ✗ | ✗ | ✗ |
| Zone TI2 | Set1 | 22 | ✗ | ✗ | ✗ | ✗ | ✗ | ✗ | ≈ |
| | Set2 | 12 | ✗ | ✗ | ✓ | ✗ | ✗ | ✗ | ✗ |
| | Set3 | 57 | ✗ | ≈ | ✓ | ✓ | ✓ | ≈ | ✓ |
| Zone TI3 | Set1 | 20 | ✗ | ✓ | ✗ | ✗ | ✗ | ✗ | ✗ |
| | Set2 | 113 | ✗ | ≈ | ✓ | ≈ | ≈ | ✗ | ✗ |
| | Set3 | 2 | ✗ | ✗ | ✗ | ≈ | ≈ | ✗ | ✗ |
| Zone TI4 | Set1 | 25 | ✗ | ✗ | ✗ | ✓ | ✓ | ≈ | ✗ |
| | Set2 | 10 | ✓ | ✓ | ✓ | ✓ | ≈ | ≈ | ≈ |
| | Set3 | 3 | ✗ | ✗ | ✗ | ✗ | ✗ | ✗ | ≈ |
| Zone TI5 | Set1 | 39 | ✓ | ≈ | ✗ | ✗ | ✗ | ✗ | ✗ |
| | Set2 | 2 | ✗ | ✗ | ✗ | ✗ | ✓ | ✓ | ≈ |
| | Set3 | 0 | ✓ | ✓ | ✓ | ✓ | ✓ | ✓ | ✓ |
| | | Satisfactory total | No | Yes | Yes | No | No | No | No |
| | | # satisfactory | 3 | 3 | 5 | 4 | 4 | 2 | 4 |
| | | # acceptable | 0 | 4 | 2 | 2 | 3 | 3 | 2 |
| | | # not acceptable | 12 | 8 | 8 | 9 | 8 | 10 | 9 |









**Table 5:** Results of the sensitivity analysis on the influence of the number of neighbours and of the variation of the acceptance threshold. The colour code is the same as the one used in table 4.

| | | Reference | SIM8 | SIM9 | SIM10 | SIM11 | SIM12 | SIM13 | SIM14 | SIM15 |
|---|---|---|---|---|---|---|---|---|---|---|
| | | | \multicolumn Number of neighbours + Acceptance threshold | | | | | | | |
| | | | \multicolumn Segments per parts | | | | | | | |
| Zone TI1 | Set1 | 156 | ✔ | ✔ | ≈ | ≈ | ✘ | ✔ | ✔ | ✔ |
| | Set2 | 95 | ✘ | ✘ | ✘ | ✘ | ✘ | ✘ | ✘ | ✘ |
| | Set3 | 6 | ✘ | ✘ | ✘ | ✘ | ✘ | ✘ | ✘ | ✘ |
| Zone TI2 | Set1 | 22 | ✘ | ✘ | ✘ | ✘ | ✘ | ✘ | ✘ | ✘ |
| | Set2 | 12 | ≈ | ≈ | ✔ | ✔ | ✘ | ✘ | ✘ | ✘ |
| | Set3 | 57 | ✔ | ✔ | ✘ | ✘ | ✔ | ✔ | ✔ | ≈ |
| Zone TI3 | Set1 | 20 | ✘ | ✘ | ✔ | ✔ | ✘ | ✘ | ✘ | ✘ |
| | Set2 | 113 | ✔ | ✔ | ≈ | ≈ | ≈ | ✔ | ✔ | ≈ |
| | Set3 | 2 | ≈ | ≈ | ✔ | ✔ | ≈ | ✘ | ✘ | ✔ |
| Zone TI4 | Set1 | 25 | ✔ | ✔ | ✘ | ✘ | ✔ | ✔ | ✔ | ✔ |
| | Set2 | 10 | ✘ | ✘ | ≈ | ≈ | ≈ | ≈ | ≈ | ✔ |
| | Set3 | 3 | ✘ | ✘ | ✘ | ✘ | ✘ | ✘ | ✘ | ✘ |
| Zone TI5 | Set1 | 39 | ✘ | ✘ | ✘ | ✘ | ✘ | ✘ | ✘ | ✘ |
| | Set2 | 2 | ≈ | ≈ | ≈ | ≈ | ✔ | ≈ | ≈ | ≈ |
| | Set3 | 0 | ✔ | ✔ | ✔ | ✔ | ✔ | ✔ | ✔ | ✔ |
| | | Satisfactory total | Yes | Yes | Yes | Yes | No | No | No | Yes |
| | | #satisfactory | 5 | 5 | 4 | 4 | 4 | 5 | 5 | 5 |
| | | #acceptable | 3 | 3 | 4 | 4 | 6 | 2 | 2 | 3 |
| | | #not acceptable | 7 | 7 | 7 | 7 | 9 | 8 | 8 | 7 |

**Table 6:** Results of the sensitivity analysis on the influence of the number of neighbours, of the variation of the acceptance threshold and of the variation of the percentage of the scanned fraction of the training image. The colour code is the same as the one used in table

| | | Reference | SIM16 | SIM17 | SIM18 | SIM19 | SIM20 | SIM21 | SIM22 | SIM23 | SIM24 | SIM25 | SIM26 | SIM27 | OPT1 |
|---|---|---|---|---|---|---|---|---|---|---|---|---|---|---|---|
| | | | | \multicolumn Number of neighbours + Acceptance threshold + % TI scan | | | | | | | | | | | Optimisation |
| | | | | \multicolumn Group 1 | | | \multicolumn Group 2 | | | | \multicolumn Group3 | | | | |
| | | | | \multicolumn Segments per parts | | | | | | | | | | | |
| Zone TI1 | Set1 | 156 | | ✔ | ✔ | ✔ | ✔ | ✘ | ✔ | ✔ | ✔ | ✘ | ✔ | ✔ | ✘ |
| | Set2 | 95 | | ✘ | ✘ | ✘ | ✘ | ≈ | ✘ | ✘ | ✘ | ≈ | ✘ | ✘ | ✔ |
| | Set3 | 6 | | ✘ | ✘ | ✘ | ✘ | ✘ | ✘ | ✘ | ✘ | ✘ | ✘ | ✘ | ✔ |
| Zone TI2 | Set1 | 22 | | ✘ | ✘ | ✘ | ✘ | ✘ | ✘ | ✘ | ✘ | ✘ | ✘ | ✘ | ✘ |
| | Set2 | 12 | | ✘ | ✘ | ✘ | ✘ | ✘ | ✘ | ✘ | ✔ | ✘ | ✔ | ✔ | ✔ |
| | Set3 | 57 | | ✔ | ✔ | ✔ | ✔ | ≈ | ✔ | ✔ | ✔ | ≈ | ✔ | ✔ | ≈ |
| Zone TI3 | Set1 | 20 | | ✘ | ✘ | ✘ | ✔ | ✘ | ✘ | ✘ | ≈ | ✘ | ✔ | ✔ | ✘ |
| | Set2 | 113 | | ✔ | ✔ | ✔ | ≈ | ✔ | ✔ | ✔ | ✔ | ✔ | ≈ | ≈ | ✔ |
| | Set3 | 2 | | ≈ | ≈ | ≈ | ✘ | ✔ | ✔ | ✔ | ≈ | ≈ | ✘ | ✘ | ✘ |
| Zone TI4 | Set1 | 25 | | ✘ | ✘ | ✘ | ≈ | ✘ | ✔ | ✔ | ✘ | ✘ | ≈ | ≈ | ✘ |
| | Set2 | 10 | | ✔ | ✔ | ✔ | ≈ | ✔ | ✔ | ✔ | ✘ | ≈ | ✔ | ✔ | ✔ |
| | Set3 | 3 | | ✘ | ✘ | ✘ | ✘ | ✘ | ✘ | ✘ | ≈ | ✘ | ✘ | ✘ | ✔ |
| Zone TI5 | Set1 | 39 | | ≈ | ≈ | ≈ | ✘ | ✘ | ✘ | ✘ | ✔ | ≈ | ✘ | ✘ | ≈ |
| | Set2 | 2 | | ≈ | ≈ | ≈ | ✘ | ✘ | ✔ | ✔ | ✔ | ≈ | ≈ | ✔ | ✔ |
| | Set3 | 0 | | ✔ | ✔ | ✔ | ✔ | ✔ | ✔ | ✔ | ✔ | ✔ | ✔ | ✔ | ✔ |
| | | Satisfactory total | | Yes | Yes | Yes | No | No | Yes | Yes | Yes | Yes | Yes | Yes | Yes |
| | | #satisfactory | | 5 | 5 | 5 | 4 | 4 | 8 | 8 | 6 | 2 | 7 | 7 | 8 |
| | | #acceptable | | 3 | 3 | 3 | 3 | 2 | 0 | 0 | 3 | 7 | 2 | 2 | 2 |
| | | #not acceptable | | 7 | 7 | 7 | 8 | 9 | 7 | 7 | 6 | 6 | 6 | 6 | 5 |

**Figure 8:** Fracture length distributions tested during the sensitivity analysis. A) fracture
length distribution for SIM1 to SIM7, B) fracture length distribution for SIM10, SIM12,
SIM13, SIM15 and C) fracture length distribution for SIM16, SIM17, SIM20, SIM21, SIM22,
SIM24, SIM5, SIM26.

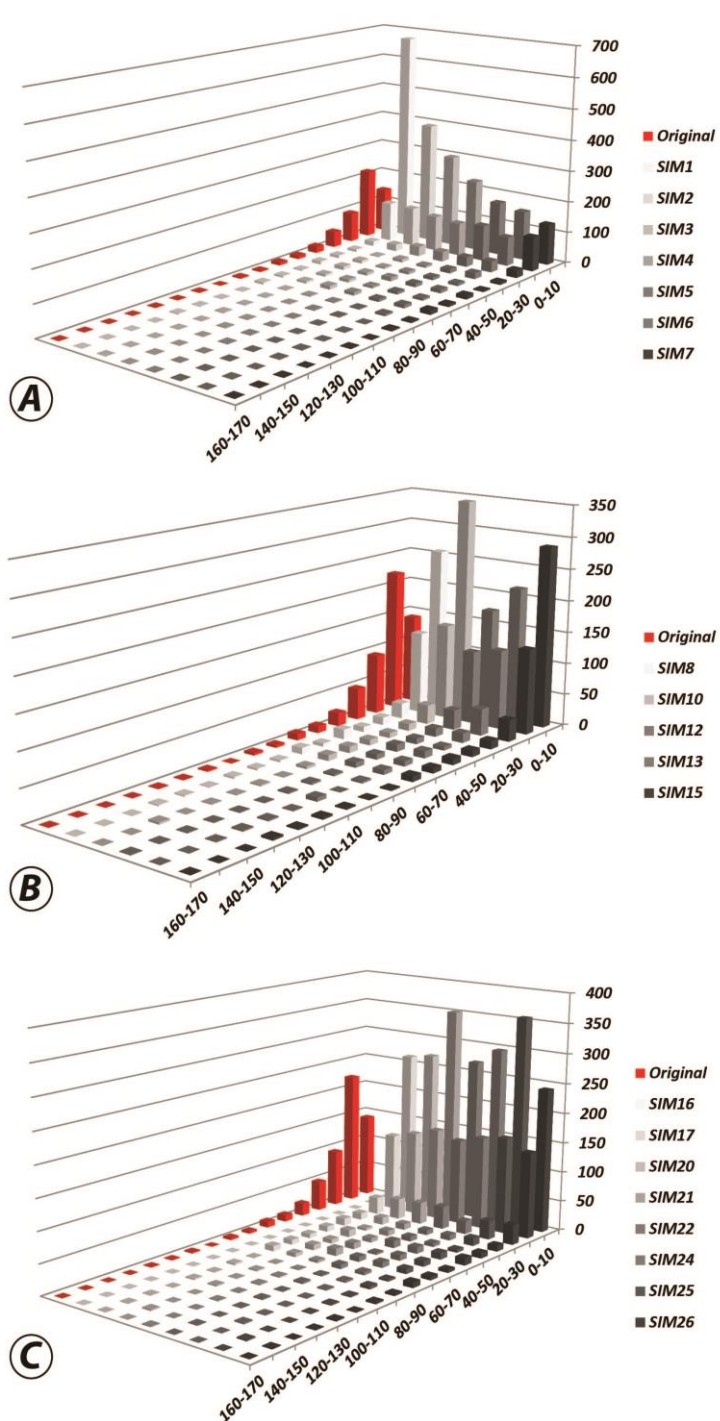


**Figure 9:** Comparison of the training images 1, 3 and 4 used during the sensitivity analysis
(27 simulations) and their modification for SIM 3

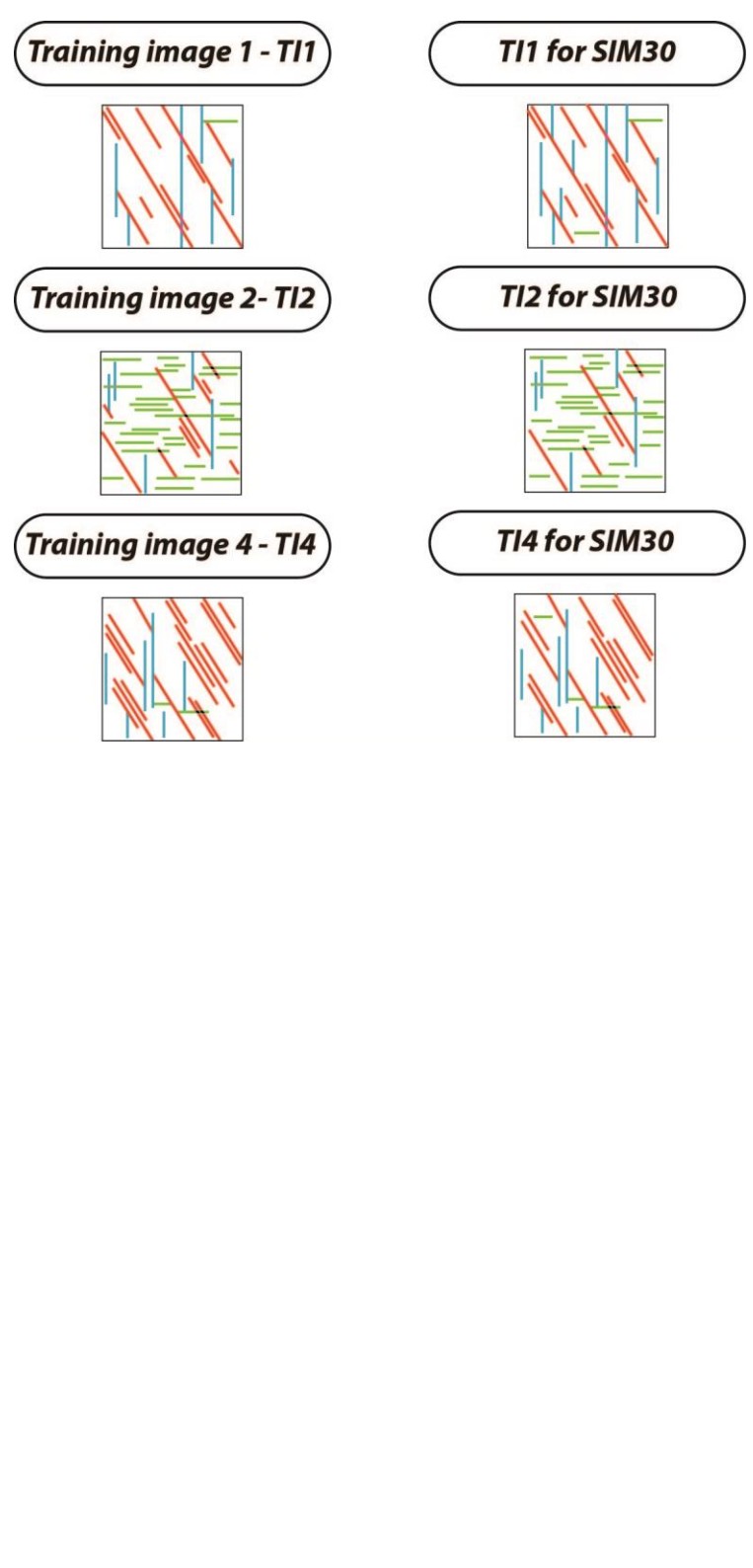















**Figure 10:** Comparison of the fracture intensity ($P_{21}$) calculated in the reference outcrop and
in four select MPS simulations

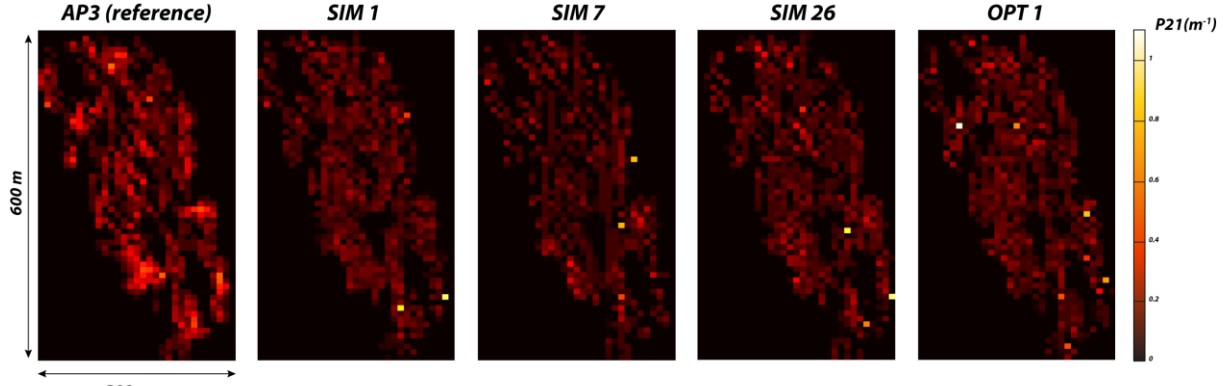

















**Figure 11:** A) Partitioning of AP4 in 3 EZ. B) probability map and associated statistics for
each EZ. C) training images associated with the partition of AP4. D) hard conditioning data
for AP4

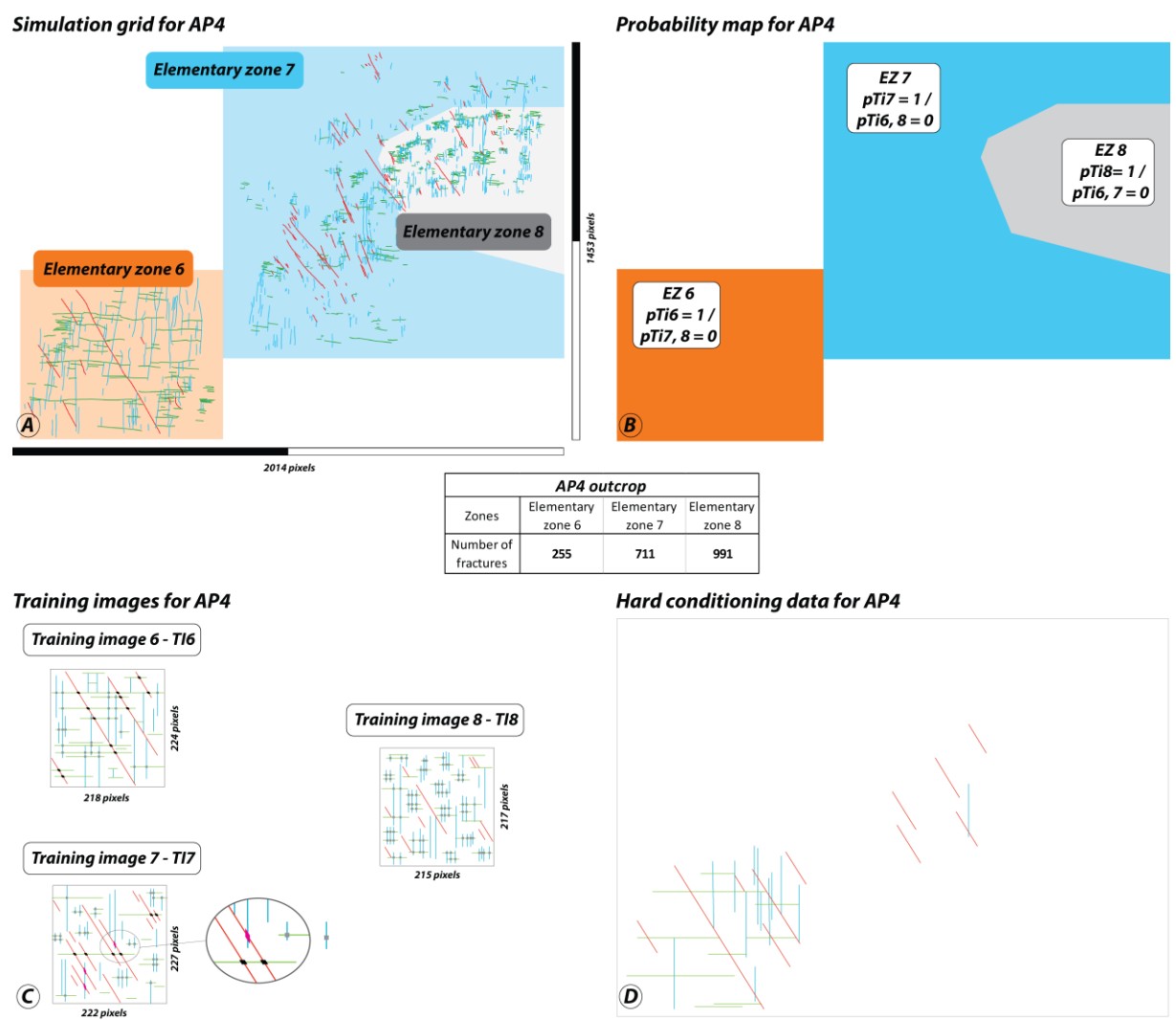

**Figure 12:** Comparison of the AP4 original outcrop with a MPS simulated version AP4-1

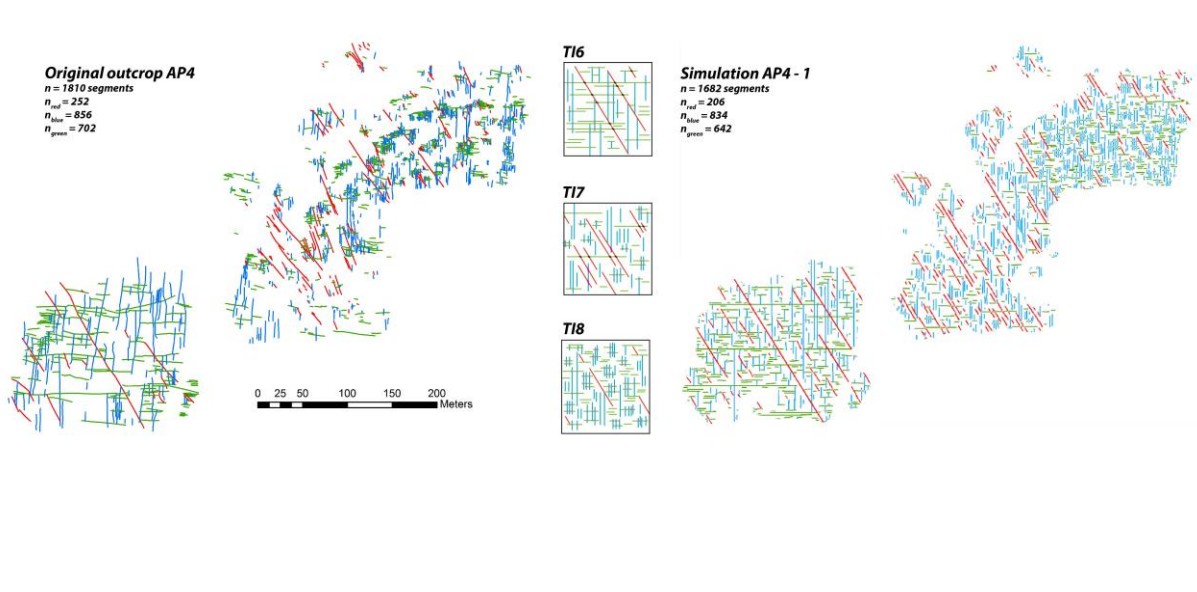




















**Figure 13:** Smooth probability map at the reservoir scale (combination of AP3 and AP4). A) Relative position of AP3 and AP4 outcrops. B) Apodi fault added into the area of interest. Extension of the probability map regions in AP3 and AP4 without geological drivers C) and with the influence of the Apodi fault D). Probability maps with smooth transition zones without geological drivers E) and with the influence of the Apodi fault F).

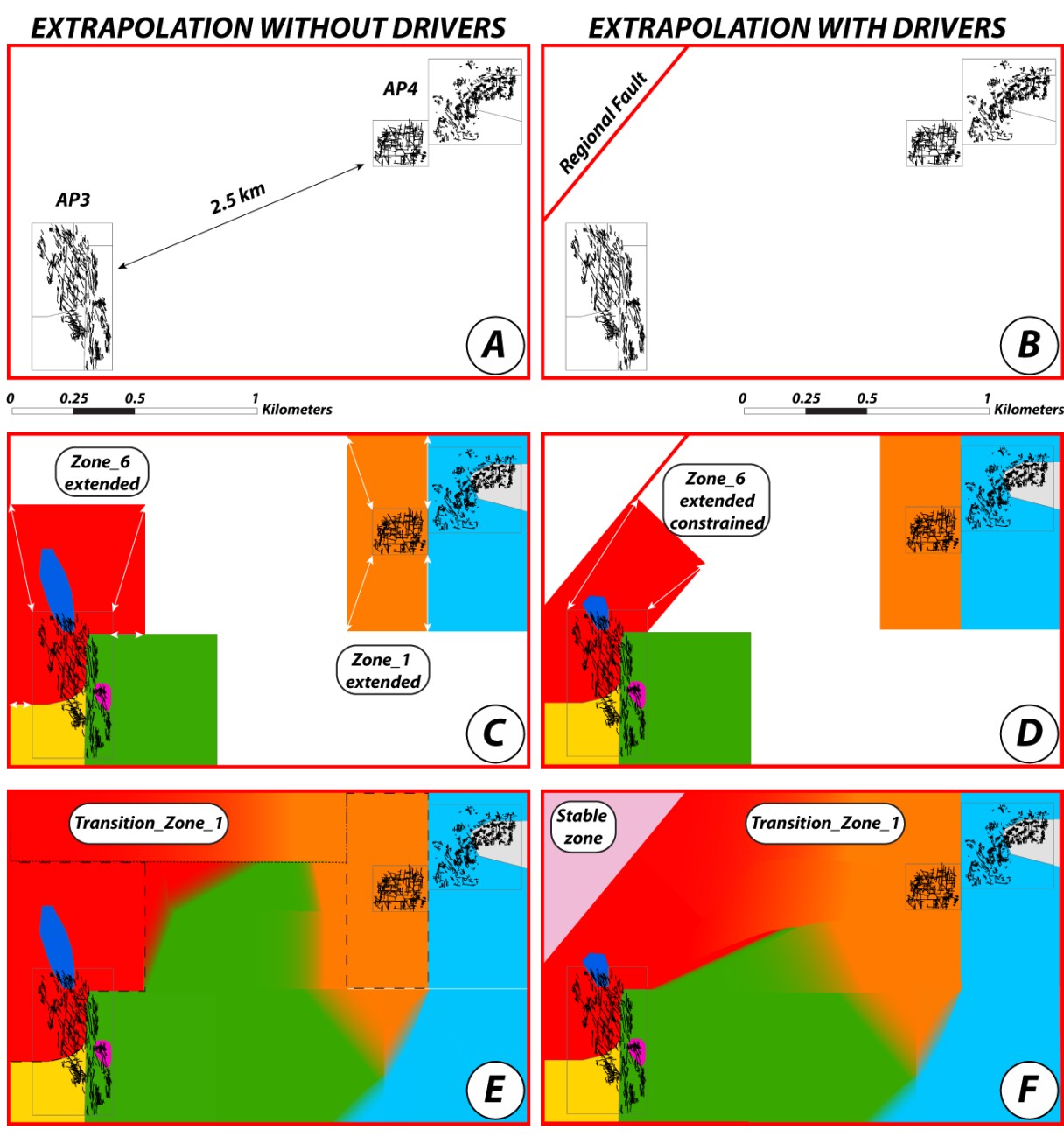

**Figure 14:** Fracture network extrusion in 3D. The method consists of identifying the different
fracture units (FU) on which the fracture height is supposed to be constant (A). This method
requires one simulation per top fracture unit (SIM SLICES). (B) is a 3D DFN based on the
hypothetical case (A) and realised in gOcad software. (C) is a cross section realised in the
centre of the 3D model in the E-W direction.

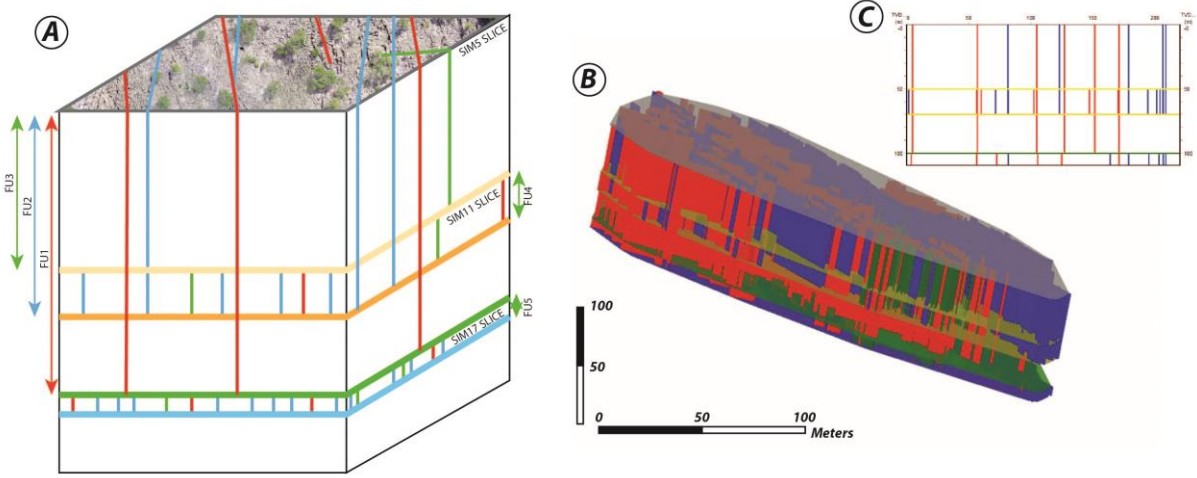

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
