# Peer review of "using Multiple Point Statistics"

_Solid Earth, 2018_

## Referee Comment (RC1) · Anonymous Referee #1 · 22 Oct 2018

The ms. by Bruna et al presents an interesting method for characterizing fracture patterns in outcrops, which involves the Multiple Point Statistics. The topic is of interest for the audience of Solid Earth, however, I had some difficulties in following many parts of the text. A robust (or at least higher than mine) background in statistics is required to fully appreciate this work. I think that, as it stands, this ms. is designed for a journal like GMD rather than for Solid Earth. I suggest either to move to another journal or to expand and explain in more detail the multiple point statistics and the training images methods.

Specific comments:

Lines 22-330. Remove from the abstract, this is material for the intro.

[Figure]

Line 30. Avoid citations in the Abstract.

Line 34. The abstract should be self supporting. Define training images

Line 35. Which process?

Line 38. Same basic concepts of statistic should be expanded.

Line 58. determined (instead of inherited).

Line 97. Explain why fracture connectivity are poorly constrained in these representations

Section II.1 (The direct sampling methods). This section is extremely difficult to follow

line 144. grid in the X&Y axis; what a node does represent?

Line 166. check "Reference"

Section II.3; sub-sub-section "Training images": this text requires a figure. It is hard to follow it.

Line 199. This does not make sense. How can an image represent a phenomenon? it is a sketch?

Line 295. How?

Line 312. I suppose that EZ are determined according to their fracture pattern

Section IV.2. Is this section really necessary for this work?

―――――――――――――――

---

## Referee Comment (RC2) · Anonymous Referee #2 · 19 Nov 2018

1) This paper presents a specific application of multipoint statistics to generate synthetic non-stationary 2-D DFN models. 2) I don't think that this is the _first_ application of MPS - I seem to remember a number of authors using e.g., SGEMS for generating conditioned fracture fields. 3) It is greatly appreciated that the paper clearly explains the limitations of the approach, particularly problems in identifying longer fractures. In addition, there are problems in 3D extrapolation, non-planar features, and termination modes. 4) In evaluating the generated DFN's, it would be appropriate to quantitatively compare the statistics of generated vs training DFN's in terms of:intensity spatial distribution as P21 m/m^2, orientation distribution for each set, and size (trace length) distribution, and termination modes . 5) I don't understand why the engineering aspects of the project ( i.e, using Barton's empirical relationship to define aperture, geomechanical

and flow simulations )- are included in the paper. This is a theoretical algorithm paper, not a case study. Perhaps the engineering portions can be put in a separate paper ? 6) I would appreciate more details on the specifics of the MPS implementation. I don't think that the algorithm could be reproduced by others with just the details provided in the paper. How is MPS used to vary fracture intensity ? trace length ? orientation ? How is MPS used to for different sets , or are the sets completely independent models ?

---

## Short Comment (SC1) · 29 Nov 2018

Overall quality: This is potentially a valuable contribution on the topic of understanding fracture networks. Outcrop fracture studies are being revolutionized by the rapid acquisition of fracture patterns from drones and photogrammetry. Developments in statistical approaches to process these observations are needed. This paper makes a credible contribution on the statistical front. And the written presentation and illustrations are fairly clear and compelling. I do think that there is room for improvement to increase the impact of the paper.

Specific comments

In the presentation encompassing figures 3 through 5, I didn't completely follow how

you defined 'fracture facies' and 'elementary zones'. Is there some sort of statistical measure of deviation from random you used (as in, for example, Marrett et al. 2018). Or are the 'facies' just qualitatively identified as 'looking similar'? My apologies if I just missed the explanation.

The Abstract reads too much like an Introduction. This part of the text needs to be more information rich. Instead of saying the paper proposes a multiple point statistics method, the Abstract should try to explain the specifics in a highly succinct way. Likewise, how was the method tested; don't just use a passive construction to tell the reader that the method 'was tested'. Bring forward some of the specifics from the Conclusions.

The Introduction could also use improvement. For one thing, the Introduction does not make a very coherent case for why outcrop studies of fractures are so essential. The reason isn't necessarily because fracture networks have 'intrinsic complexity' (line 65)—some networks are quite simple—but because the elements of fracture patterns that govern fluid flow, like connectivity and height and length distribution and the apparent clustered distributions evident in figs 3-5 cannot be adequately sampled in the subsurface. Some attributes like length distribution cannot be sampled at all in the subsurface. Outcrops are where these features can be measured. The Introduction would be stronger if it spelled out this challenge in clear, simple terms.

It would also help if the cited literature included some more explicit examples of how these hard- or impossible-to-measure attributes affect fluid flow (for example, Long & Witherspoon 1985 on connectivity; Olson et al. 2009 on length distribution in unconnected networks in porous rocks). Right now the Introduction 'lacks motivation'. Many of the parts are there but the case needs to be made stronger. See some of the specific comments below.

Ok; the following might seem like a tangential issue. But generalist readers need to have a clear explanation of what problems there might be in using outcrop fracture

[Figure]

patterns as analogs for those in the subsurface. In section I.2 about surface rocks as reservoir analogs, an incautious reader would never suspect from the text here that there might be problems with using outcrops fractures for this purpose. This omission needs to be fixed. Some outcrop fractures provide close matches to those in subsurface areas of interest (e.g., Gomez-Rivas et al., 2014) but others do not (e.g., Laubach et al., 2009). In many cases, outcrop fractures provide demonstrably misleading guidance for the subsurface (Corbett et al., 1987 and subsequent work on the Austin Chalk cited in Laubach et al. 2009; Li et al., 2018).

Studies typically seek to omit fractures that result from near-surface processes unrelated to fractures at depth (Stearns & Friedman, 1972). But subsurface sampling over the past two decades shows that in the moderate- to deep subsurface (1 km+) in sedimentary basins, many fracture pattern elements differ from those found in more readily sampled outcrops even if the fractures in those outcrops formed in the subsurface, and for unsurprising reasons. Comparative studies in the same rock type and structural setting of fracture spacing observed in outcrop and sampled in long fracture-perpendicular cores shows that patterns in exposures can differ markedly from those in the nearby subsurface (Li et al., 2018, J. Struct. Geol.). The differing temperature-pressure paths of outcrops and rocks at depth and associated differences in rock properties are key reasons that the evidence outcrop patterns provide on fracture patterns in the deeper subsurface needs to be used with caution. The need for caution should be mentioned even if this particular outcrop is a good subsurface analog.

Part of the process of using outcrop fractures is figuring out to what extent the outcrops are guides, and to what circumstances, of the subsurface. This part of the Introduction should acknowledge this issue and mention that the authors addressed it (I notice that later in the MS the outcrops are said to be good analogs; can the authors mention why?).

I'm sure the authors recognize this issue and despite the length of my comments a brief but complete acknowledgment of the issue is all that is needed in my opinion.

The statistical approach seems like a reasonable one. But I think the paper would benefit from a clearer explanation perhaps aimed at a generalist audience, as well as featuring a compare-and-contrast with other similar approaches. I noticed that the Liu et al. 2002 citation in your reference list is incomplete. [Liu, X., Srinivasan, S., & Wong, D. (2002, January). Geological characterization of naturally fractured reservoirs using multiple point geostatistics. In SPE/DOE Improved Oil Recovery Symposium. Society of Petroleum Engineers.] If you go to OnePetro you can get the doi for papers like this one.

I'd be interested in seeing a comparison with the Hanke et al 2018 directional semivariogram (J. Struct. Geol. 108 [March]).

I didn't find the analysis of aperture variation to really be much of a test and the whole exercise seems a bit extraneous to the statistical analysis of the pattern. The text needs to explain more clearly in what sense this is a test (even if that turns out to underline that it is a limited test). As noted below, it would also be appropriate to present the 'stress sensitivity' (or not) of fractures in a more nuanced way. Why no direct measurements of aperture size distributions?

Technical questions & comments

30 Abstracts do not normally contain citations.

53 'Ubiquitous' means that fractures are everywhere but excavations and horizontal core studies show that some rocks in the subsurface lack fractures, or if fractures are present they are so widely spaced (hundreds of meters or more) that 'everywhere' is not an apt description. An outcrop example showing how resistant to fracture some rocks are is Ellis et al. 2012, J. Geol. Soc. London. A better word might be 'widespread'. Moreover, areas of completely sealed fractures are also common in the subsurface, and such fractures are rarely fluid conduits. Although I don't agree with people who don't count such rocks as fractured, it's certainly the case that some rocks lack fracture flow conduits.

55 I think more caution is called for in citing for this point (effects of fractures on fluid flow). There are relatively few papers that document the effects of fractures on fluid flow in hydrocarbon reservoirs but many papers that repeat the contention that fractures are important for fluid flow. One of the papers that does quantify production data with respect to natural fractures is Solano et al, 2011 SPE Reservoir Evaluation & Engineering. However, although both of the papers cited here in the MS are interesting contributions, I don't think they are the right papers to cite in support of the point the authors make. All of the references mentioned in his section of the text should be reviewed with this point in mind.

58-ca. 62 Ok, so maybe a quibble, but 'well known'? really? Maybe I'm not following what the authors are trying to say here, but connecting the specific strain and stress conditions to the formation of a given fracture or fracture pattern is full of uncertainty: the timing of fracture formation is commonly very challenging to estimate unambiguously and because fracture arrays are generally low strain phenomena and through geologic time a wide range of loading paths might lead to fracture (e.g., Engelder 1985, J. Struct. Geol.) the connection between pattern and cause is frequently ambiguous. A good example relevant to this paper is fractures in outcrop. Did they form due to some process at depth (for example, elevated pore fluid pressure) or during uplift or exposure? This issue gets to the reliability of outcrop-derived fracture pattern information (which I'm all in favor of obtaining) but the challenge of determining the causes of fractures I think needs a bit more thoughtful or nuanced treatment.

67 Do you mean stresses in the past when fracture patterns formed (paleo stresses)? You seem to be claiming that fractures are highly sensitive to current stress state. I know this is a widely accepted premise, but you should at least note that many reservoirs are known to have fractures that are stiff and insensitive to current stress state (e.g., Laubach et al., 2004, Earth & Planetary Science letters).

71-86 This section needs to contain some caveats about the limitations of outcrop fracture research.

73 The use of outcrop fracture patterns to constrain the subsurface goes much deeper into the past than the recent references cited here: National Research Council 1996. Rock fractures and fluid flow: Contemporary understanding and applications. National Academy Press, Washington D.C., 551 p.

81-82 The 'how, when, and where' is rarely obvious from the pattern alone. Flagging this comment is not off topic since it relates to how outcrop data can or should be used.

89 'provide'

93-94 This sounds like jargon; provide a clearer explanation of what you mean for a general audience.

113-119 This is too late in the MS to introduce this material. Some of this could be in the Abstract.

125 What do you mean by 'full outcrops'. This seems vague. If you have a size range in mind, why not state it?

135 I'm not sure I follow you here. You didn't measure any apertures in outcrop, did you? So is this just a process of a computation applied to both the outcrop imaged fractures and the statistical realizations? Why no measured outcrop apertures?

205-206 Some of the text here sounds like it is carry over from a proposal, since you've done the work.

271 Does the karst figure into your aperture calculations?

322 This seems late in the text to have this kind of preview of goals?

365 Interesting. Are some of the >40-m-long fractures still censored by outcrop size?

572-575 There are some jumps in logic here. Yes, flow depends on open fractures. But whether or not fractures are open or not does not simply depend on in situ stress conditions. Some (many) fractures are insensitive to stress state (they are very stiff) and

some are closed because they are mineral filled. It therefore does not necessarily follow that 'contribution of fractures to fluid flow. . .can be defined by the Mohr-Coulomb. . .' etc. The development here needs to be more nuanced and include a few caveats.

It is also worth noting I think that the predominant role of aperture in fluid flow presumes a completely impermeable host rock, which is generally not a good assumption even for low porosity unconventional reservoirs (TGS; shale). If there is flow in the host rock and the fractures are not interconnected, open length distribution is what matters (Philip et al. 2005). Philip et al. varied the apertures in their simulations by a lot and got no significant difference in flow. Philip, Z. G., et al., 2005, Modeling coupled fracture-matrix fluid flow in geomechanically simulated fracture networks: SPE Reservoir Evaluation & Engineering, 8/4, 300-309.

576 'a key parameter'; if it's a key parameter, why were apertures not measured in the field?

622 'statistic'; is this the word you mean? Obscure usage.

625 What do you mean by 'aborted' fractures? Non-standard usage; suggest you pick another word.

632 Mechanical stratigraphy is readily measured in the subsurface; 'fracture stratigraphy' is more challenging.

Did you rigorously describe your fracture height patterns for the outcrops (maybe it is in one of the cited references). Height patterns and fracture stratigraphies have different patterns. There is a useful classification in Hooker et al. 2013, J. Struct. Geol.

637 'fracture family' is non-standard usage. Is there a reason not to call these groupings 'fracture sets' (Hancock, 1985)?

641 'provides'; ('The method provides a realistic. . .')

Fig. 8, caption 'Fracture. . .'

Fig. 10. Some of the colors on this figure make it hard to read.

Fig. 12 would be more informative with more labeling and explanation on the face of the figure. Add a graphic explanation/key.

---

## Author Response (AR1)

**A new methodology to train fracture network simulation using Multiple Point Statistics**

Pierre-Olivier BRUNA[(1)*], Julien STRAUBHAAR[(2)], Rahul PRABHAKARAN[(1, 3)], Giovanni BERTOTTI[(1)], Kevin BISDOM[(4)], Grégoire MARIETHOZ[(5)], Marco MEDA[(6)].

(1) Department of Geoscience and Engineering, Delft University of Technology, Delft, the Netherlands.
(2) Centre d'hydrogéologie et de géothermie (CHYN), Université de Neuchâtel, Emile-Argand 11, CH-2000 Neuchâtel.
(3) Department of Mechanical Engineering, Section of Energy Technology, Eindhoven University of Technology, Eindhoven the Netherlands.
(4) Shell Global Solutions International B.V., Grasweg 31, 1031HWAmsterdam, The Netherlands
(5) University of Lausanne, Institute of Earth Surface Dynamics (IDYST) UNIL-Mouline, Geopolis, office 3337, 1015 Lausanne, Switzerland
(6) ENI Spa, Upstream and Technical Services, San Donato Milanese, Italy.

* Corresponding author, p.b.r.bruna@tudelft.nl

Keywords: geostatistics, multiple training images, probability map, fracture networks, stress-induced fracture aperture, outcrop.

**Abstract**

Natural fracture network characteristics can be known from high-resolution outcrop images acquired from drone and photogrammetry. These outcrops might also be good analogues of subsurface naturally fractured reservoirs and can be used to make predictions of the fracture geometry and efficiency at depth. However, even when supplementing fractured reservoir models with outcrop data, gaps in that model will remain and fracture network extrapolation methods are required. In this paper we used fracture networks interpreted in two outcrops from the Apodi area in Brazil to present a revised and innovative method of fracture network geometry prediction using the Multiple Point Statistics (MPS) method.

The MPS method presented in this article uses a series of small synthetic training images (TIs) representing the geological variability of fracture parameters observed locally in the field. The TIs contain the statistical characteristics of the network (i.e. orientation, spacing, length/height and topology) and allow representing complex arrangement of fracture networks. These images are flexible as they can be simply sketched by the user.

We proposed to use simultaneously a set of training images in specific elementary zones of the Apodi outcrops defined in a probability map in order to best replicate the non-stationarity of the reference network. A sensitivity analysis was conducted to emphasize the influence of the conditioning data, the simulation parameters and the used training images. Fracture density computations were performed on the best realisations and compared to the reference outcrop fracture interpretation to qualitatively evaluate the accuracy of our simulations. The method proposed here is adaptable in terms of training images and probability map to ensure the geological complexity is accounted for in the simulation process. It can be used on any type of rock containing natural fractures in any kind of tectonic context. This workflow can also be applied to the subsurface to predict the fracture arrangement and fluid flow efficiency in water, heat or hydrocarbon fractured reservoirs.

**I] Introduction**

**I.1 The importance of the prediction of fracture network geometry**

 Fracture are widespread in Nature and depending on their density and their aperture, they might have a strong impact on fluid flow and fluid storage in water (Berkowitz, 2002; Rzonca, 2008), heat (Montanari et al., 2017; Wang et al., 2016) and hydrocarbon reservoirs (Agar and Geiger, 2015; Lamarche et al., 2017; Solano et al., 2010) They are typically organised as networks ranging from nanometre to multi-kilometre scale (Zhang, 2016), and present systematic geometrical characteristics (i.e. type, orientation, size, chronology, topology) that are determined from specific stress and strain conditions. These conditions are typically well known These conditions have been used to derive concepts of fracture arrangements in various tectonic contexts and introduced the notion of geological fracture-drivers (fault, fold, burial, facies). Based on these drivers it is possible to some extent to predict reservoir heterogeneity in folded (Suppe, 1983, 1985; Tavani et al., 2015; Watkins et al., 2017), faulted (Faulkner et al., 2010; Matonti et al., 2012; Micarelli et al., 2006) and burial related contexts (Bertotti et al., 2017; Bisdom, 2016) and have been used to derive concepts of fracture arrangements. These concepts are currently used and to define potential permeability pathways within the rock mass (Lamarche et al., 2017; Laubach et al., 2018). Despite the existence of these concepts, fracture networks generally present an intrinsic complexity (e.g. variability of orientation, local occurrence of new fracture family, change of topology relationships) due, among others, to local variability of the orientation and magnitude of stresses. This makes fractures hard to predict. Moreover, fractures in subsurface reservoirs are largely unknown due to their sub-seismic size and to the scarcity of available data, which is generally limited to borehole information. 
[revised manuscript text omitted]

      **IV.2 Evaluation of the accuracy of the simulation using mechanical**

**modelling**

Uncertainty analysis is required while performing simulations of geological parameters, especially far from input data. The sensitivity analysis presented in this paper is a way to appreciate how good the result provided by the MPS simulation is compared to the reference.

However, this test has little meaning when the reference is unknown (e.g. subsurface).

In this case the evaluation of fracture network permeability (i.e. which of the considered fractures are open and are potentially capable to conduct fluid from one point to another) can be tested dynamically and appears to be a good way to validate the provided simulation. This approach can also be extended to fluid flow simulations through the permeable fractures. We calculated mechanical and hydraulic apertures using a Barton-Bandis stress induced aperture model (Barton, 1982; Barton and Bandis, 1980) on the original AP3 interpreted outcrop and on four selected MPS realisations.

Flow in naturally fractured reservoirs is driven by the occurrence of open fractures. The contribution of fractures to fluid flow at in-situ stress conditions in reservoir models can be defined by the Mohr-Coulomb critical stress method (Hoek and Martin, 2014; Rogers, 2003).

If shear stress acting on a fracture exceeds normal stress it becomes critically stressed and forms a conduit to flow. The fracture aperture is a key parameters to know for reservoir evaluation and production. The Barton-Bandis empirical model quantifies the aperture that remains when irregular mismatching fracture walls are partially closed under in-situ stress (Fig. 13). The theory and equations behind the use of the Barton Bandis model are presented in the appendix A of this article.

We chose four MPS generated simulations and the original interpreted fracture dataset for the aperture calculations. We used ABAQUS (Dassault Systemes), a commercial finite element solver for the stress calculations. A conformal mesh consisting of triangular elements is

$$Sh_{max} = 30MPa \qquad \nu = 0.3\,(-) \qquad JRC = 15\,(-)$$
$$Sh_{min} = 10MPa \qquad E = 35\,GPa \qquad JCS = 120MPa$$

[revised manuscript text omitted]

**Appendix A**

In the Barton-Bandis stress aperture model, the hydraulic fracture aperture is a function of local stresses, shear displacement, initial roughness of the fracture and mechanical properties of the rock. A set of empirical functions for the mechanical aperture was defined by (Barton, 1982). (Olsson and Barton, 2001) defined the hydraulic aperture as a function of the mechanical aperture. The initial mechanical aperture is given by the following relation from which the mechanical aperture is a function of the Joint Roughness Coefficient (JRC), Joint Compressive Strength (JCS) and uniaxial compressive strength (in MPa).

$$E_0 = \frac{JRC}{5}\left(0.2\frac{\sigma_c}{JCS} - 0.1\right)$$

The JRC is a measure of the relative roughness of the rock fracture surface and is commonly measured using a Barton comb. The JCS can be measured in the field using Schmidt hammer rebound measurements. The joint compressive strength (in MPa) is a mechanical property of the rock. For our simulations, we set JRC = 15 and JCS equal to the uniaxial compressive strength of 120 MPa, so that we obtain an initial unstressed mechanical aperture of 0.3 mm which is independent of length or network geometry. This is valid for an un-weathered fracture where the initial unstressed aperture is only a function of roughness. The mechanical aperture is a function of the normal stress, initial stiffness and the maximum closure.

$$E_n = E_0 - \left(\frac{1}{v_m} + \frac{K_{ni}}{\sigma_n}\right)^{-1}$$

The initial stiffness (in MPa/mm) and maximum closure (in mm) are functions of the JRC, JCS and the initial mechanical aperture.

$$K_{ni} = (-7.15) + 1.75 JRC + 0.02\left(\frac{JCS}{E_0}\right)$$

$$v_m = A + B\,(JRC) + C\left(\frac{JCS}{E_0}\right)^D$$

$$
e =
\begin{cases}
for \ \dfrac{u_s}{u_{peak}} \leq 0.75 \ ; \dfrac{E_n^{\ 2}}{JRC^{2.5}} \\[2em]
for \ \dfrac{u_s}{u_{peak}} \geq 1; \sqrt{E_n} \ JRC_{mob}
\end{cases}
$$

——————————————————

$$
u_{\text{peak}} = 0.0077 L^{0.45} \left( \frac{\sigma_n}{\text{JCS}} \right)^{0.34} \cos \left( \text{JRClog}_{10} \left[ \frac{\text{JCS}}{\sigma_n} \right] \right)
$$

——————————————

$$
JRC_{mob} = JRC \left[ \frac{u_s}{u_{peak}} \right]^{-0.381}
$$

[revised manuscript text omitted]

Van Eijk, M.: Analysis of the fracture network in carbonate rocks of the Jandaira Formation in
northeast Brazil, 2014. Technische Universiteit Delft, 60 pp., 2014.

Vollgger, S. A. and Cruden, A. R.: Mapping folds and fractures in basement and cover rocks
using UAV photogrammetry, Cape Liptrap and Cape Paterson, Victoria, Australia, Journal of
Structural Geology, 85, 168-187, 2016.

Wang, S., Huang, Z., Wu, Y.-S., Winterfeld, P. H., and Zerpa, L. E.: A semi-analytical
correlation of thermal-hydraulic-mechanical behavior of fractures and its application to
modeling reservoir scale cold water injection problems in enhanced geothermal reservoirs,
Geothermics, 64, 81-95, 2016.

Watkins, H., Healy, D., Bond, C. E., and Butler, R. W. H.: Implications of heterogeneous
fracture distribution on reservoir quality; an analogue from the Torridon Group sandstone,
Moine Thrust Belt, NW Scotland, Journal of Structural Geology, doi:
https://doi.org/10.1016/j.jsg.2017.06.002, 2017. 2017.

Wu, J., Boucher, A., and Zhang, T.: A SGeMS code for pattern simulation of continuous and
categorical variables: FILTERSIM, Computers & Geosciences, 34, 1863-1876, 2008.

Zhang, L., Kang, Q., Chen, L., Yao, J.: Simulation of flow in multi-scale porous media using the
lattice boltzmann method on quadtree grids, Communications in Computational Physics 19,
17, 2016.

Dear Reviewer

First let us thank you very much for your comments and for the interest you found in our paper.

**In the first paragraph of your review you stated: "this manuscript is designed for a journal like GMD rather than Solid Earth".**

As you also mentioned earlier in the review, we believe that the topic is relevant for Solid Earth as it presents an innovative method to build DFN models considering more geology than the existing (exclusively statistical or almost) methods. As stated in the manuscript, we are using an existing algorithm from Mariethoz et al., 2010 that we simply tuned and applied to fracture network geometry predictions. We also think that the paper is quite long and developing the MPS method and the training image further will probably not be really helpful for the reader. We also cited a series of reference papers explaining more in detail how the DS algorithm was written (Mariethoz et al., 2010), how could be used the DS algorithm (Meershmann et al., 2013) and presenting a series of application of MPS to generate fracture networks (Chugunova et al., 2017, Karimpouli et al., 2017, Jung et al., 2013). We believe that these references will greatly help readers to go deeper into the MPS method.

In the manuscript no changes are required
* * *
Specific comments
**Line 22-33, remove from the abstract, this is material for the intro.**

We agree with the reviewer comment and we believe that the initial part of the abstract need to be rephrased . As the same material is stated (differently) into the introduction we did not reintegrated this paragraph there.

In the manuscript, we rewrote line 22 to line 30 from the abstract as following: Natural fractures have a strong impact on flow and storage properties of reservoirs. Their distribution in the subsurface is largely unknown mainly due to their sub-seismic scale and to the scarcity of available data sampling them (borehole). Outcrop can be considered as analogues where natural fracture characteristics can be extracted from high-resolution images acquired from drone and photogrammetry. Outcrops thus become a digital laboratory where the interpreted fracture network can be tested mechanically (fracture aperture, distribution of strain/stress) and dynamically (fluid flow simulations).

**Line 30 Avoid citations in the Abstract**:

We agree with the reviewer comment

As this line is part of the previously removed paragraph, this comment is already taken into account

**Line 34 The abstract should be self-supporting. Define training images**:

A training image is a sketch drawn by the user and containing patterns (series of pixels) representing a geological object (e.g. a fluvial channel or a fracture network). In this case, all the statistics generated during the simulation process are not dependent on the simulation algorithm but come from the user-designed training image (Journel., 2003).

Consequently, the realisations should represent a geological concept the user has in mind.
The concept of training image has been fairly well defined in previous studies and we are not sure that this definition will be beneficial to our article.

In the manuscript, we did not extended further the definition of the training image as we thought that the concept is familiar to the major part of the audience of this paper

**Line 35 Which process?:**

we agree that a process cannot be reproduced by an image. In fact it is a sketch of a network supposed to be representative of the network.

In the manuscript, the term was removed from the manuscript.

**Line 38 Same basic concepts of statistic should be expanded:**

we understand there that you would like us to define the non-stationarity. As stated in line 93-94: "the local stationarity hypothesis suggests the invariance of all of the generated statistics by translation in the simulated domain". This means for instance that if the user decide that the spacing of fracture family X is 1 m, then each X fracture will be spaced accordingly in the simulation domain. In our case we want to create sub domains, intrinsically stationary but overall inducing variability in the simulation grid. In our case Family X in domain Y (elementary zone) will have a spacing of 1m but family X in domain Z (elementary zone) will have a spacing of 2m.
* * *
In the Manuscript the abstract was entirely revised as following
Natural fractures have a strong impact on flow and storage properties of reservoirs. Their distribution in the subsurface is largely unknown mainly due to their sub-seismic scale and to the scarcity of available data sampling them (borehole). Outcrop can be considered as analogues where natural fracture characteristics can be extracted from high-resolution images acquired from drone and photogrammetry. Outcrops thus become a digital laboratory where the interpreted fracture network can be tested mechanically (fracture aperture, distribution of strain/stress) and dynamically (fluid flow simulations). One of those outcrop, a flat pavement from the Apodi area in Brazil, was used as a benchmark to evaluate how good are the Multiple Point Statistics (MPS) to replicate the complex arrangement of a reference manually-interpreted fracture network. The MPS method presented in this article is innovative as it is based on the creation of small and synthetic training images representing the variability of the distribution of fracture parameters observed in the field. These images are flexible as they can be simply sketched by the user. We proposed to use simultaneously a set of training images in specific elementary zones defined in a probability map in order to best represent the non-stationarity of the reference network. A sensitivity analysis emphasizing the influence of the conditioning data, the simulation parameters and the used training images was conducted on the obtained simulations. Fracturing density computations and stress-induced fracture aperture calculations were performed on the best realisations and compared the reference outcrop fracture interpretation to qualitatively evaluate the accuracy of our simulations. The method proposed here is adaptable 5in term of training images and probability map) and introduces geology since the initial part of the simulation process. It can be used on any type of rocks containing natural fractures in any kind of tectonic context. This workflow can also be applied to the subsurface to predict the fracture arrangement and its fluid flow efficiency in water, heat or hydrocarbon reservoirs.

**Line 58 determined (instead of inherited):**

modification taken into account in the manuscript

**Line 97 Explain why fracture connectivity are poorly constrained in these representations:**

The fracture network connectivity implies crosscutting or abutting relationship. While scanning the training image, the MPS algorithm is looking for patterns. In this case abutments are easily found, as it is a singular combination of pixel color. However a crosscutting relationship implies that one fracture is above another. Pixel-wise there will be one fracture continuous and another one discontinuous as the crossing locus cannot be the two colors at the same time. Then without considering a particular facies at the crosscutting locus the fact that two fractures are crossing each other is not taken into account and consequently the connectivity is (partially) lost.

In the manuscript we did not explained more as this problem is detailed in line 325-347 and associated with a figure (6)

**Section II.1 (The direct sampling methods). This section is extremely difficult to follow**

We are aware that this section is difficult to understand as it implies some terms that are very specific to MPS method. However we tried to make a resume of Mariethoz et al., 2010 and others authors that used the same method before us. This part is also very short and we think that it is not mandatory to understand the rest of the paper. We believe however that it is necessary to present this part to audience that is more aware of MPS technique for them to understand on which method we base our calculations.

In the manuscript we decided to not change this part as it is already simplified and must be included into the paper in our sense. However we added in supplementary material a pseudocode of the DS algorithm to help people to better understand how DS is working.

**Line 144. Grid in the X&Y axis; what a node does represent?**
A node represents a pixel of the grid. A node in the simulation grid is called x, whereas a node in the training image grid is called y. Here "x" and "y" are not related to an axis of the grid.

No change seems to be required in the manuscript

**Line 166. Check "Reference"**
We apologize for this error into the manuscript. It should have been removed. Indeed it will not appear in the revised manuscript

In the manuscript we removed this error

**Section II.3; sub-sub-section "Training images": this text requires a figure. It is hard to follow it.**
We do not believe that a figure of what is a training image should be required. The training image is nothing complex but a simple sketch drawn by the geologist. In the case of fracture network it could also be a photograph with an interpretation on it (representative element area for instance) used to populate one part of the simulation domain.

In the manuscript we will cite figures 5, 6, 9 and 10 where the training images appear to make them more explicit to the reader.

**Line 199. This does not make sense. How can an image represent a phenomenon? It is a sketch?**

See above the reply concerning line 35

**Line 295. How?**

We agree that this sentence is out of place

We removed this sentence from the manuscript

**Line 312. I suppose that EZ are determined according to their fracture pattern**

Yes they are. This is what we meant by visual inspection of the pavement.

We do not think that action is required in the manuscript for this issue

**Section IV.2. Is this section really necessary for this work?**

The reviewer is probably not fully aware of reservoir simulation grid resolutions and the need for an upscaled value of permeability. One of the purposes of our MPS technique is to generate DFN realizations of which the local response is later upscaled to a field scale reservoir model. For small-scale fracture networks as ours, the fracture aperture is an unknown quantity, which greatly affects the local pressure response. The matter in Section IV.2 discusses a method to calculate this unknown and compare the aperture variability between various MPS realizations. The results of such a geomechanical approach highlight the robustness of the MPS method as it is able to recreate networks that have similar
regions where apertures are open / closed and hence the implications are that they would have a similar fluid flow pressure response.

However following similar comments from the other reviewer we decided to remove this part from the manuscript

Dear Reviewer

First let us thank you very much for the interesting and constructive comments you left on our paper. This letter aims to reply to your specific comments concerning our manuscript

**1) This paper presents a specific application of multipoint statistics to generate synthetic non-stationary 2-D DFN models. I don't think that this is the _first_ application of MPS - I seem to remember a number of authors using e.g., SGEMS for generating conditioned fracture fields.**

As far as we were aware of we were the first proposing a MPS approach using the Direct Sampling algorithm were multiple training images are used at time in order to predict the geometry of fracture networks in a non-stationary manner. We also developed our own Matlab codes to generate probability maps or to extract segments out of a MPS realisation (image). We are aware about the fact that people in the past tried to model fracture networks using MPS method however they used single training image that should represent the whole variability of the network. After looking specifically for similar work associated to the laboratory developing SGEMs we were unfortunately not able to find any results suggesting that our approach was already tested. If the reviewer can provide specific references we will be very pleased to take them into consideration and to eventually change the title of our article.

We do not think that particular action in the manuscript are required from the reviewer concerning this issue

**2) It is greatly appreciated that the paper clearly explains the limitations of the approach, particularly problems in identifying longer fractures. In addition, there are problems in 3D extrapolation, non-planar features, and termination modes.**

Thank you for this comment. Indeed we do have some limitations however we are working on methods for solving some of the problems you are mentioning and we hope to be able to release them as publications in the near future.

We do not think that particular action in the manuscript are required from the reviewer concerning this issue

**3) In evaluating the generated DFN's, it would be appropriate to quantitatively compare the statistics of generated vs training DFN's in terms of: Intensity spatial distribution as P21 m/mˆ2, orientation distribution for each set, and size (trace length) distribution, and termination modes.**

In this paper we made a sensitivity analysis comparing the reference data (fracture tracing made by a geologist on high-resolution drone imagery) with the realisations we obtained. In this part of the work we compared the amount of segments and their length distribution to evaluate if the realisations are good enough or not. We also used a non-standard approach using the Barton-Bandis stress induced aperture calculation to compare some of the best realisations with the reference outcrop.
While, we think that all of those tests are sufficient to show that the approach is giving satisfactory results in this specific case,

We agree to add to our revised manuscript some P21 calculation conducted on the 5 selected realisations presented in figure 14. This part will be inserted just before the section IV on uncertainties

**4) I don't understand why the engineering aspects of the project (i.e, using Barton's empirical relationship to define aperture, geomechanical and flow simulations)- are included in the paper. This is a theoretical algorithm paper, not a case study. Perhaps the engineering portions can be put in a separate paper**?

Indeed, this attempt of validation of our MPS results is not a conventional way of doing it. However, we strongly believe that calculating fracture aperture has a double interest. Intrinsically the method allows obtaining realistic estimations of fracture apertures in a non-stationary fracture network. These kind of data cannot be obtained directly from observation in the field and getting these data is of primary importance for fluid flow calculations. Secondly, fracture aperture in Apodi outcrops were already calculated by Bisdom et al., 2016, and we wanted to use this work as a base to compare the results of our simulation. By doing so we demonstrated that reference and trained networks behave equally. Indeed they provide similar ranges of apertures and tend to locate open fractures in the same areas. We believe that this approach is an original and interesting way to validate the results of the MPS.

However following similar comments from the other reviewer we decided to remove this part from the manuscript

**5) I would appreciate more details on the specifics of the MPS implementation. I don't think that the algorithm could be reproduced by others with just the details provided in the paper. How is MPS used to vary fracture intensity? Trace length? Orientation? How is MPS used to for different sets, or are the sets completely independent models?**

As previously mentioned by Journel, 2003 or by Strebelle 2002, the key factor in MPS simulation is the training image used to generate the model. This is why some authors like Hu et al., 2014 use a full reservoir model as training image. In that case all of the heterogeneity is considered and the statistical variability is implicitly generated – among others – during the scanning of the training image. The idea of our paper is to use fracture facies corresponding to the different sets of fractures identified in an entire outcrop – in our case 3 – and to use multiple sketches to vary the parameters you are talking about: the intensity, the length of fracture the crosscutting relationship and more. If you check into figure 5 for instance, you will see that the training images carry a lot of implicit information gathered from field based investigations for instance or from the interpretation of the network from the drone image. The result obtained showed that the algorithm is able to reproduce this variability during the MPS process. We thought that this part is well explained in the "Method" part under "Multiscale fracture attributes" and "Training images, conditioning data and probability map" chapter. However if we do need to rewrite this paragraph we will be pleased to do it on the revised version of our manuscript.

We decided to add a piece of pseudocode in the manuscript as supplementary material to help people to be able to better understand what DS is doing. The following pseudocode is suggested

The DeeSse algorithm (Straubhaar, 2011) was used in this paper to reproduce existing fracture network interpreted from outcrop pavements. The following pseudocode developed by Oriani et al., 2017 have been modified to explains how the algorithm is processing the simulation of fracture. Specific terms can be found in section II.1 of the present paper. In our study the simulation follow a random path into the simulation grid. This grid is step by step populated by pixels y sampled in the training image until the simulation grid is entirely filled by properties called **V(x)** (fracture facies in this case). The algorithm proceeds according to the following sequence

1. Selection of a random location **x** in the simulation grid that has not yet been simulated (far from any conditioning data already inserted in the grid)

2. To simulate **V(x)** → the fracture facies into the simulation grid: retrieve a data event $d_n(x)$, corresponding to **n** neighbours around **x** thanks to a fixed circular spatial window of radius R. The pattern $d_n(x) = (x_1, V(x_1)),...,(x_n, V(x_n))$ formed by at most **n** informed nodes the closest to **x** is retrieved. If no neighbours is assigned (at the beginning of the simulation) and $d_n(x)$ will then be empty: In this case, assign the value **V(y)** of a random location y to **V(x)**, and repeat the procedure from the beginning.

3. Visit a random location y in the TI and retrieve the corresponding data event $d_n(y)$.

4. Compare $dn(x)$ to **dn(y)** using a distance **D($dn(x)$, $dn(y)$)** corresponding to a measure of dissimilarity between the two data events.

5. If **D($dn(x)$, $dn(y)$)** is smaller than a user-defined acceptance threshold **T**, the value of **V(y)** is assigned to **V(x)**. Otherwise step 3 to step 5 are repeated until the value is assigned or an given fraction of the TI, **F** is scanned.

6. if **F** is scanned, **V(x)** are defined as the scanned datum that minimise the distance **D($dn(x)$, $dn(y)$)** within the simulation grid.

7. Repeat the whole procedure until all the simulation grid is informed.

Dear Stephen Laubach

First let us thank you very much for the very detailed and highly valuable comments you left on our paper. In this letter we tried to reply to all of your comments as precisely as possible. We hope that our answer will satisfy you.

**Overall quality: This is potentially a valuable contribution on the topic of understanding fracture networks. Outcrop fracture studies are being revolutionized by the rapid acquisition of fracture patterns from drones and photogrammetry. Developments in statistical approaches to process these observations are needed. This paper makes a credible contribution on the statistical front. And the written presentation and illustrations are fairly clear and compelling. I do think that there is room for improvement to increase the impact of the paper.**

**Specific comments**
**C1:In the presentation encompassing figures 3 through 5, I didn't completely follow how you defined 'fracture facies' and 'elementary zones'. Is there some sort of statistical measure of deviation from random you used (as in, for example, Marrett et al. 2018). Or are the 'facies' just qualitatively identified as 'looking similar'? My apologies if I just missed the explanation**.

We decided arbitrarily that the facies would be defined in regard of fracture sets (based on orientation). The facies can eventually be made more complex if the user wants for instance to separate different length of fracture within the same set of fracture.
The elementary zones are based on the variability of fracturing intensity per set within the outcrop. This analysis is possible because we have access to the final network that we consider as the "reality". In this case, when we observed a drastic change in the network geometry we placed a boundary around this area and this boundary defines an elementary zone. For instance, EZ1 contains mainly the NS (blue) and the NW-SE (red) fracture sets. On the contrary EZ2 contains mainly EW (green) fractures and EZ3 mainly NS fractures. EZ4 and 5 represent patches where the fracture density is higher. You can see the new density maps in the revised manuscript.

In the manuscript this is explained first in part II.3, probability map. We there the following paragraph
The PM comes from a simple sketch (i.e. a pixelated image) given by the MPS user. It is based on the geological concepts or interpretations that define the geometry variability over the simulated area and that allow a partition of the outcrop. In each of the zones defined into the area of interest, the simulated property will follow the intrinsic stationary hypothesis (citation) but the entire domain will be non-stationary.
While working on outcrops, the partition of the area of interest can be determined based on observations. For instance, when the fracture network interpreted from outcrop images is available, the geologist can visually define where the characteristics of the network are changing (fracture orientation, intensity, length, topology) and draw limits around zones where the network remains the same. This technique was used in the present paper. However, outcrops or subsurface may lack of continuity between observation sites. If the data are sparse and come mainly from fieldwork ground observation or boreholes, the use of alternative statistical approaches can help to provide a robust and accurate partition of the area of interest. The work of Marett et al., (2018) interprets the spatial organisation of fractures using advanced statistic techniques such as normalized correlation count and weighted correlations count, on scanlines collected in the Pennsylvanian Marble Falls Limestone in the United States. In their approach, the periodicity of fracture spacing (clustering) calculated from the mentioned techniques is evaluated using Monte Carlo quantifying how different from a random organisation are arranged the fractures in the investigated network. These approaches can be highly valuable during the process of building a probability maps when less data are available. The probability maps provide a large-scale....

**C2: The abstract reads too much like an Introduction. This part of the text needs to be more information rich. Instead of saying the paper proposes a multiple point statistics method, the Abstract should try to explain the specifics in a highly succinct way. Likewise, how was the method tested; don't just use a passive construction to tell the reader that the method 'was tested'. Bring forward some of the specifics from the Conclusions.**

We agree on that point.

In the manuscript, the abstract was modified as following:
Natural fracture network characteristics can be known from high-resolution outcrop images acquired from drone and photogrammetry. These outcrops might also be good analogues of subsurface naturally fractured reservoirs and can be used to make predictions of the fracture geometry and efficiency at depth. However, even when supplementing fractured reservoir models with outcrop data, gaps in that model will remain and fracture network extrapolation methods are required. In this paper we used fracture networks interpreted in two outcrops from the Apodi area in Brazil to present a revised and innovative method of fracture network geometry prediction using the Multiple Point Statistics (MPS) method.
The MPS method presented in this article uses a series of small synthetic training images (TI's) representing the geological variability of fracture parameters observed locally in the field. The TI's contain the statistical characteristics of the network (i.e. orientation, spacing, length/height and topology) and allow representing complex arrangement of fracture networks. These images are flexible as they can be simply sketched by the user.
We proposed to use simultaneously a set of training images in specific elementary zones of the Apodi outcrops defined in a probability map in order to best replicate the non-stationarity of the reference network. A sensitivity analysis was conducted to emphasize the influence of the conditioning data, the simulation parameters and the used training images. Fracture density computations were performed on the best realisations and compared to the reference outcrop fracture interpretation to qualitatively evaluate the accuracy of our simulations. The method proposed here is adaptable in terms of training images and probability map and ensure the geological complexity is accounted for in the simulation process. It can be used on any type of rock containing natural fractures in any kind of tectonic context. This workflow can also be applied to the subsurface to predict the fracture arrangement and fluid flow efficiency in water, heat or hydrocarbon fractured reservoirs.

**C3: The introduction could also use improvement. For one thing, the Introduction does not make a very coherent case for why outcrop studies of fractures are so essential. The reason isn't necessarily because fracture networks have 'intrinsic complexity' (line 65) "some networks are quite simple" but because the elements of fracture patterns that govern fluid flow, like connectivity and height and length distribution and the apparent clustered distributions evident in figs 3-5 cannot be adequately sampled in the subsurface. Some attributes like length distribution cannot be sampled at all in the subsurface. Outcrops are where these features can be measured. The Introduction would be stronger if it spelled out this challenge in clear, simple terms.**
**It would also help if the cited literature included some more explicit examples of how these hard- or impossible-to-measure attributes affect fluid flow (for**

**example, Long & Witherspoon 1985 on connectivity; Olson et al. 2009 on length distribution in unconnected networks in porous rocks). Right now the Introduction 'lacks motivation'. Many of the parts are there but the case needs to be made stronger. See some of the specific comments below.**

We thank you very much for this comment. We will modify the manuscript accordingly.

In the manuscript
Line 65: Despite the existence of these concepts, a range of parameters including fracture abutment relationships as well as height/length distributions cannot be adequately sampled along a 1D borehole and are mainly invisible on seismic images. In addition, fracture networks may present a spatial complexity (variability of orientation or clustering effect) that is also largely unknown in the subsurface. Long and Witherspoon (1985) and Olson et al., (2009) showed how those parameters impact the connectivity of the network and consequently affect fluid flow in the subsurface. In outcrops the fracture network characteristics can be observed and understood directly. Consequently outcrops are essential to characterize fracture network attributes that cannot be sampled in the subsurface, such as length or spatial connectivity.

**C4:Ok; the following might seem like a tangential issue. But generalist readers need to have a clear explanation of what problems there might be in using outcrop fracture patterns as analogs for those in the subsurface. In section I.2 about surface rocks as reservoir analogs, an incautious reader would never suspect from the text here that there might be problems with using outcrops fractures for this purpose. This omission needs to be fixed. Some outcrop fractures provide close matches to those in subsurface areas of interest (e.g., Gomez-Rivas et al., 2014) but others do not (e.g., Laubach et al., 2009). In many cases, outcrop fractures provide demonstrably misleading guidance for the subsurface (Corbett et al., 1987 and subsequent work on the Austin Chalk cited in Laubach et al. 2009; Li et al., 2018). Studies typically seek to omit fractures that result from near-surface processes unrelated to fractures at depth (Stearns & Friedman, 1972). But subsurface sampling over the past two decades shows that in the moderate- to deep subsurface (1 km+) in sedimentary basins, many fracture pattern elements differ from those found in more readily sampled outcrops even if the fractures in those outcrops formed in the subsurface, and for unsurprising reasons. Comparative studies in the same rock type and structural setting of fracture spacing observed in outcrop and sampled in long fracture-perpendicular cores shows that patterns in exposures can differ markedly from those in the nearby subsurface (Li et al., 2018, J. Struct. Geol.). The differing temperature-pressure paths of outcrops and rocks at depth and associated differences in rock properties are key reasons that the evidence outcrop patterns provide on fracture patterns in the deeper subsurface needs to be used with caution. The need for caution should be mentioned even if this particular outcrop is a good subsurface analog. Part of the process of using outcrop fractures is figuring out to what extent the outcrops are guides, and to what circumstances, of the subsurface. This part of the Introduction should acknowledge this issue and mention that the authors addressed it (I notice that later in the MS the outcrops are said to be good analogs; can the authors mention why?). I'm sure the authors recognize this issue and despite the length of my comments a brief but complete acknowledgment of the issue is all that is needed in my opinion.**

We will also follow this advice as we are sharing the same opinion on analogues. The issue pointed in the second paragraph of this comment is also approached in the revised text proposed below.

In the manuscript after the line 86 the following text was added

However, not every outcrops can be considered as good analogues for the subsurface. Li et al., (2018), in their work on the Upper Cretaceous Frontier Formation reservoir, USA observed significant differences in the fracture network arrangement in subsurface cores compared to an apparent good surface analogue of the studied reservoir. In the subsurface, fractures appear more clustered than in the outcrop where the arrangement is undistinguishable from random. The origin of these difference is still debated but these authors suggest that alteration (diagenesis) or local change in pressure-temperature conditions, may have conducted to the observed variability. The near-surface alteration processes (exhumation, weathering) may also conduct to misinterpretations of the characteristics of the network. In this case, one should be particularly carful while using observed networks to make geometry or efficiency (porosity, permeability) predictions in the subsurface. Therefore, the application of the characteristics observed in the outcrop to the subsurface is not always straightforward or even possible, and may lead to erroneous interpretations. Relatively unbiased signals such as stylolites or veins and particular geometric patterns might be a trustful basis to show that the studied surface fracture can be, to some extends, compared to the subsurface.

**C5:The statistical approach seems like a reasonable one. But I think the paper would benefit from a clearer explanation perhaps aimed at a generalist audience, as well as featuring a compare-and-contrast with other similar approaches. I'd be interested in seeing a comparison with the Hanke et al 2018 directional semivariogram (J. Struct. Geol. 108 [March]).**
**I noticed that the Liu et al. 2002 citation in your reference list is incomplete. [Liu, X., Srinivasan, S., & Wong, D. (2002, January). Geological characterization of naturally fractured reservoirs using multiple point geostatistics. In SPE/DOE Improved Oil Recovery Symposium. Society of Petroleum Engineers.] If you go to One Petro you can get the doi for papers like this one.**

It is true that a lot of work has been done in geostatistics concerning the simulation of fracture network. For instance the work of Bruna et al., (2015 JOH) using two points statistic to evaluate the connectivity of fractured geobodies, the extensive literature using simple or sophisticated DFN (Fracman-type approach) or the approach you mentioned from Hanke et al., 2018 (appearing very interesting). However we believe that MPS is a bit apart in terms of algorithm but also in term of implementation. As we already stated in the new abstract, the goal here is to integrate more geology from scratch (even before starting the simulation) in a series of simple images which are in that respect much more flexible than an average value of density along a well for instance. This is why we did not extend too much on the comparison with other existing methods.

In the manuscript we believe that the new abstract and the part I.3 are sufficient to give to the reader a hint of what are the approaches classically used or used in the past and to present how different and flexible is the MPS approach. However, we added the following paragraph after line 106:

Work of Hanke et al., (2018) uses a directional semivariogram to quantify fracture intensity variability and intersection density. This contribution provides an interesting way to evaluate the outputs of classical DFN approaches but require a large quantity of input data that are not always available in the subsurface. An alternative geologically-constrained method which i) explicitly predict the organisation and the characteristics of multiscale fracture objects, ii) takes into consideration the spatial variability of the network and iii) requires a limited amount of data to be realised would be an interesting and innovative way to represent fracture network geometry in various contexts.
The second paragraph of the comment was modified in the reference list.

**C6:I didn't find the analysis of aperture variation to really be much of a test and the whole exercise seems a bit extraneous to the statistical analysis of the pattern. The text needs to explain more clearly in what sense this is a test (even if that turns out to underline that it is a limited test). As noted below, it would also be appropriate to present the 'stress sensitivity' (or not) of fractures in a more nuanced way. Why no direct measurements of aperture size distributions?**

We had similar comments from one of the other reviewer of our paper and we agreed to remove the part talking about fracture aperture IV.2. We admit that the position of this part is not adequately positioned and that the test has a limited impact on the validation of the method already given by a detailed sensitivity analysis.  Concerning your last comment we did not took into consideration direct measurements of aperture size distribution for the same reason you mentioned earlier in the review (C4 alteration process).  In fact veins are not available everywhere and in each considered sets. This is why the modelling approach appeared to us more robust.

In the manuscript, as already proposed in the answer to one of our Anonymous Reviewer we will add a small visual comparison of the P21 between the 5 models proposed in the figure 14.  This part will be added as a III.5 before the discussion.
* * *
Technical questions & comments
**C7:30 Abstracts do not normally contain citations.**

The citation will be removed

In the manuscript the new abstract will not contain citations.

**C8:53 'Ubiquitous' means that fractures are everywhere but excavations and horizontal core studies show that some rocks in the subsurface lack fractures, or if fractures are present they are so widely spaced (hundreds of meters or more) that 'everywhere' is not an apt description. An outcrop example showing how resistant to fracture some rocks are is Ellis et al. 2012, J. Geol. Soc. London. A better word might be 'widespread'. Moreover, areas of completely sealed fractures are also common in the subsurface, and such fractures are rarely fluid conduits. Although I don't agree with people who don't count such rocks as fractured, it's certainly the case that some rocks lack fracture flow conduits.**

We understand this comment and we will modify the phasing in the manuscript.

In the manuscript the sentence will be modified as following: Fracture are widespread in Nature and depending on their density and their aperture, they might have a strong impact on fluid flow and fluid storage in....

**C9:55 I think more caution is called for in citing for this point (effects of fractures on fluid flow). There are relatively few papers that document the effects of fractures on fluid flow in hydrocarbon reservoirs but many papers that repeat the contention that fractures are important for fluid flow. One of the papers that does quantify production data with respect to natural fractures is Solano et al, 2011**

**SPE Reservoir Evaluation & Engineering. However, although both of the papers cited here in the MS are interesting contributions, I don't think they are the right papers to cite in support of the point the authors make. All of the references mentioned in his section of the text should be reviewed with this point in mind.**

We think that the authors cited in this section are all dealing with fractures affecting subsurface reservoirs or surface reservoir-analogues in different contexts and they all seem to converge on the conclusion that fracture play a role (positive or negative) in fluid flow. We agree however that the paper from Solano deserves to be cited there in addition to the Agar and Geiger and Lamarche et al.

In the Manuscript the reference was added in line 55

**C10:58-ca. 62 Ok, so maybe a quibble, but 'well known'? really? Maybe I'm not following what the authors are trying to say here, but connecting the specific strain and stress conditions to the formation of a given fracture or fracture pattern is full of uncertainty: the timing of fracture formation is commonly very challenging to estimate unambiguously and because fracture arrays are generally low strain phenomena and through geologic time a wide range of loading paths might lead to fracture (e.g., Engelder 1985, J. Struct. Geol.) the connection between pattern and cause is frequently ambiguous. A good example relevant to this paper is fractures in outcrop. Did they form due to some process at depth (for example, elevated pore fluid pressure) or during uplift or exposure? This issue gets to the reliability of outcrop-derived fracture pattern information (which I'm all in favor of obtaining) but the challenge of determining the causes of fractures I think needs a bit more thoughtful or nuanced treatment.**

We admit that a strong shortcut has been made there and that the phrasing has to be revised. In our case fractures form in the response of loading. The differences in patterns between two outcrops AP3 and AP4 distant of about 2.5 kilometres are still debated. However Bertotti et al., 2017 bring some new answer to those questions.

In the manuscript the lines 58-62 have been removed and replaced by: These conditions have been used to derive concepts of fracture arrangements in various tectonic contexts and introduced the notion of geological fracture-drivers (fault, fold, burial, facies). Based on these drivers it is possible to some extents to predict reservoir heterogeneity…

**C12:67 Do you mean stresses in the past when fracture patterns formed (paleo stresses)? You seem to be claiming that fractures are highly sensitive to current stress state. I know this is a widely accepted premise, but you should at least note that many reservoirs are known to have fractures that are stiff and insensitive to current stress state (e.g., Laubach et al., 2004, Earth & Planetary Science letters).**

We were there talking about paleostress. However, this paragraph was modified according to the comment you made earlier.

In the manuscript: see the response provided in C3

**C13: 71-86 This section needs to contain some caveats about the limitations of outcrop fracture research.**

This issue was addressed in the general comments mentioned before and have been modified in the manuscript.

In the manuscript: see the response provided in C3

**C14: 73 The use of outcrop fracture patterns to constrain the subsurface goes much deeper into the past than the recent references cited here: National Research Council 1996. Rock fractures and fluid flow: Contemporary understanding and applications. National Academy Press, Washington D.C., 551 p.**

We agree that this reference is important and we will follow the advice of the reviewer without removing most recent citations.

In the manuscript: the reference was added

**C15:81-82 The 'how, when, and where' is rarely obvious from the pattern alone. Flagging this comment is not off topic since it relates to how outcrop data can or should be used.**

This was the goal of this sentence. A lot of interpretations is possible from collected outcrop data. This is why we put the word eventually in this sentence. We do not see specially what the reviewer means there.

In the manuscript we did not changed this sentence.

**C16:89 'provide'**

Changed

In the manuscript we modified "are providing" with "provide"

C17:93-94 This sounds like jargon; provide a clearer explanation of what you mean for a general audience.

We modified the text

In the manuscript the sentence was separated in two parts:
The generated models follow a local stationarity hypothesis. This implies that the statistics used during the simulation are constant in the defined area of interest….

**C18:113-119 This is too late in the MS to introduce this material. Some of this could be in the Abstract.**

We agreed on that and we changed the abstract as per the answer to comment C2

In the manuscript the abstract was changed

**C19:125 What do you mean by 'full outcrops'. This seems vague. If you have a size range in mind, why not state it?**

We were working on outcrops which sizes are in the order of 100m. The exact dimension of these outcrops is presented in table1.

In the manuscript the sentence was modified as following:
"…geometry variability over outcrops (size order of $10^2$m) and a methodology."

**C20:135 I'm not sure I follow you here. You didn't measure any apertures in outcrop, did you? So is this just a process of a computation applied to both the outcrop imaged fractures and the statistical realizations? Why no measured outcrop apertures?**

Some apertures were measured in outcrops but we did not used them in this work as we favoured the modelling part. We did not took into consideration surface fracture apertures because they were not representative of the subsurface conditions (weathering issues and exhumations). In any case the part on fracture aperture will be removed from the manuscript.

In the manuscript the sentence: "we computed mechanical and hydraulic apertures in outcrop fracture interpretation and on the obtained stochastic models." Was removed and replaced by "we computed density maps in outcrop fracture interpretation and on selected stochastic models."

**C21:205-206 Some of the text here sounds like it is carry over from a proposal, since you've done the work.**

We will remove "propose to" from the manuscript

In the manuscript we replaced it by "we used multiple training image"

**C22:271 Does the karst figure into your aperture calculations?**

Unfortunately not. But we believe that the karstification is due to "recent" surface alteration and will not be present in the subsurface as we can see them today in the outcrop. This topic is close to the one discussed in C4.

In the manuscript: no change applied except that the part on aperture was removed.

**C23:322 This seems late in the text to have this kind of preview of goals?**

We agree that this sentence is out of place

In the manuscript we removed line 332 to 324.

**C24:365 Interesting. Are some of the >40-m-long fractures still censored by outcrop size?**

Yes few of these fractures are censored by the boundaries of the outcrop. However a large majority of the fracture are included inside the pavement and we assume that they are representative of the maximal length of the fractures there.

No changes were requested in the manuscript

**C25:572-575 There are some jumps in logic here. Yes, flow depends on open fractures. But whether or not fractures are open or not does not simply depend on in situ stress conditions. Some (many) fractures are insensitive to stress state (they are very stiff) and some are closed because they are mineral filled. It therefore does not necessarily follow that 'contribution of fractures to fluid flow: : :can be defined by the Mohr-Coulomb: : :' etc. The development here needs to be more nuanced and include a few caveats. It is also worth noting I think that the predominant role of aperture in fluid flow presumes a completely impermeable**

**host rock, which is generally not a good assumption even for low porosity unconventional reservoirs (TGS; shale). If there is flow in the host rock and the fractures are not interconnected, open length distribution is what matters (Philip et al. 2005). Philip et al. varied the apertures in their simulations by a lot and got no significant difference in flow. Philip, Z. G., et al., 2005, Modelling coupled fracture-matrix**
**fluid flow in geomechanically simulated fracture networks: SPE Reservoir Evaluation &**
**Engineering, 8/4, 300-309.**

Thank you for this comment and for the interesting reference you provided. However as this part was removed from the paper this matter will (hopefully) be addressed in a separate paper.

In the manuscript part IV.2 was removed

**C26:576 'a key parameter'; if it's a key parameter, why were apertures not measured in the field?**

As discussed previously we wanted to apply the aperture calculation into subsurface conditions so we did not considered fracture aperture measured in the field.

In the manuscript part IV.2 was removed

**C27:622 'statistic'; is this the word you mean? Obscure usage**.

We agree on this comment

In the manuscript the sentence was replaced by: "To Tackle these problems we choose to use multiple 2D MPS-generated fracture networks".

**C28:625 What do you mean by 'aborted' fractures? Non-standard usage; suggest you pick another word.**

We agree on this comment

In the manuscript the word aborted was removed

**C29:632 Mechanical stratigraphy is readily measured in the subsurface; 'fracture stratigraphy' is more challenging. Did you rigorously describe your fracture height patterns for the outcrops (maybe it is in one of the cited references). Height patterns and fracture stratigraphies have different patterns. There is a useful classification in Hooker et al. 2013, J. Struct. Geol.**

In fact the aim of this part was to provide a way to use 2D MPS in 3D. We showed our idea and we built a very simple 3D DFN based on our outcrop. The method is now much more elaborated and takes into consideration the issue you mention.

In the manuscript we had the following sentence after the Laubach et al., 2009 citation: The fracture height distribution, refered as fracture stratigraphy (Hooker et al., 2013) requires here a particular attention and is difficult to extract from borehole data. In outcrops, the use of vertical cliffs adjacent to 2D horizontal pavement should be a way to evaluate these height and to constrain the 3D model.

**C30:637 'fracture family' is non-standard usage. Is there a reason not to call these groupings 'fracture sets' (Hancock, 1985)?**

We agree on this comment

In the manuscript fracture family was replaced by fracture sets

**C31:641 'provides'; ('The method provides a realistic: : :')**

We agree on this comment

In the manuscript the suggested sentence was inserted

**C32:Fig. 8, caption 'Fracture: : :'**

We agree on this comment

In the manuscript fracture will appears with a capital "F" there

**C33:Fig. 10. Some of the colours on this figure make it hard to read.**

We agree on this comment

The figure was modified for the article

[Figure]

**C34:Fig. 12 would be more informative with more labelling and explanation on the face of the figure. Add a graphic explanation/key.**

We agree on this comment

The figure was modified for the article

[Figure]

**EXTRAPOLATION WITHOUT DRIVERS**

**EXTRAPOLATION WITH DRIVERS**

AP4

AP3

2.5 km

A

Regional Fault

B

0,125  0,25  0,5  0,75  1  Kilometers

0,125  0,25  0,5  0,75  1  Kilometers

Zone_6 extended

Zone_1 extended

C

Zone_6 extended constrained

D

Transition_Zone_1

E

Stable zone

Transition_Zone_1

F

---

## Referee Report (RR1)

17, 2016.

[referee-annotated manuscript omitted]